# A calcium transport mechanism for atrial fibrillation in Tbx5-mutant mice

Wenli Dai[1†], Brigitte Laforest[2†], Leonid Tyan[1], Kaitlyn M Shen[2], Rangarajan D Nadadur[2], Francisco J Alvarado[3], Stefan R Mazurek[4], Sonja Lazarevic[2], Margaret Gadek[2], Yitang Wang[1], Ye Li[1], Hector H Valdivia[3], Le Shen[1,5], Michael T Broman[4], Ivan P Moskowitz[2]*, Christopher R Weber[1]*

[1]Department of Pathology, University of Chicago, Chicago, United States; [2]Departments of Pediatrics, Pathology, and Human Genetics, University of Chicago, Chicago, United States; [3]Department of Medicine, Division of Cardiovascular Medicine, University of Wisconsin-Madison School of Medicine and Public Health, Madison, United States; [4]Department of Medicine, University of Chicago, Chicago, United States; [5]Section of Neurosurgery, Department of Surgery, University of Chicago, Chicago, United States

*For correspondence:
imoskowitz@peds.bsd.uchicago.edu (IPM);
christopher.weber@uchospitals.edu (CRW)

[†]These authors contributed equally to this work

**Abstract** Risk for Atrial Fibrillation (AF), the most common human arrhythmia, has a major genetic component. The T-box transcription factor TBX5 influences human AF risk, and adult-specific *Tbx5*-mutant mice demonstrate spontaneous AF. We report that TBX5 is critical for cellular $Ca^{2+}$ homeostasis, providing a molecular mechanism underlying the genetic implication of TBX5 in AF. We show that cardiomyocyte action potential (AP) abnormalities in *Tbx5*-deficient atrial cardiomyocytes are caused by a decreased sarcoplasmic reticulum (SR) $Ca^{2+}$ ATPase (SERCA2)-mediated SR calcium uptake which was balanced by enhanced trans-sarcolemmal calcium fluxes (calcium current and sodium/calcium exchanger), providing mechanisms for triggered activity. The AP defects, cardiomyocyte ectopy, and AF caused by TBX5 deficiency were rescued by phospholamban removal, which normalized SERCA function. These results directly link transcriptional control of SERCA2 activity, depressed SR $Ca^{2+}$ sequestration, enhanced trans-sarcolemmal calcium fluxes, and AF, establishing a mechanism underlying the genetic basis for a $Ca^{2+}$-dependent pathway for AF risk.
DOI: https://doi.org/10.7554/eLife.41814.001

## Introduction

Atrial fibrillation (AF) is the most common arrhythmia in humans, characterized by irregularly irregular atrial electrical activity, resulting in asynchronous atrial contraction. AF is a global problem, affecting more than 33 million people and approximately 25% of Americans over the age of forty (*Nishida and Nattel, 2014*; *Weng et al., 2018*). AF is associated with significant morbidity and mortality due to thromboembolic events, heart failure, and sudden cardiac death. AF also significantly complicates overall health care management, with AF patients costing five times more to treat than patients without AF (*Andrade et al., 2014*). The total annual cost to treat AF patients in the US is on the order of 26 billion dollars (*Nishida and Nattel, 2014*). AF is a highly significant and growing public health concern.

A genetic basis for AF risk has been described in the last decade. Large community-based cohort studies indicate that heritability provides between 40% and 62% of AF risk (*Nishida and Nattel, 2014*; *Christophersen et al., 2009*). An emerging paradigm describes AF as a multifactorial disease with genetic predisposition that will determine the propensity of secondary clinical insults to cause AF. This model highlights the importance of understanding the molecular mechanisms underlying

**eLife digest** The human heart contains four distinct chambers that work together to pump blood around the body. In individuals with a condition called atrial fibrillation, two of the chambers (known as the atria) beat irregularly and are unable to push all the blood they hold into the other two chambers of the heart. This can cause heart failure and increases the likelihood of blood clots, which may lead to stroke and heart attacks.

Small molecules called calcium ions play a crucial role in regulating how and when the atria contract by driving electrical activity in heart cells. To contract the atria, a storage compartment within heart cells known as the sarcoplasmic reticulum releases calcium ions into the main compartment of the cells. Calcium ions also enter the cell from the surrounding tissue. As the atria relax, calcium ions are pumped back into the sarcoplasmic reticulum or out of the cell by specific transport proteins.

Individuals with mutations in a gene called *Tbx5* are more likely to develop atrial fibrillation than other people, but it was not clear how such gene mutations contribute to the disease. Here, Dai, Laforest et al. used mice with a mutation in the *Tbx5* gene to study how defects in *Tbx5* affect electrical activity in heart cells.

The experiments found that the *Tbx5* gene was critical for calcium ions to drive normal electrical activity in mouse heart cells. Compared with heart cells from normal mice, the heart cells from the mutant mice had decreased flow of calcium ions into the sarcoplasmic reticulum and increased flow of calcium ions out of the cell.

These findings provide a direct link between atrial fibrillation and the flow of calcium ions in heart cells. Together with previous work, these findings indicate that multiple different mechanisms could lead to atrial fibrillation, but that many of these involve changes in the flow of calcium ions. Therefore, personalized medicine, where clinicians uncover the specific mechanisms responsible for atrial fibrillation in individual patients, may play an important role in treating this condition in the future.

DOI: https://doi.org/10.7554/eLife.41814.002

the genetic predisposition to AF. Genome-wide association studies (GWAS) studies have identified common risk variants and familial mutations at the T-box transcription factor 5 (TBX5) locus that result in increased risk for AF (*McDermott et al., 2008*; *Sinner et al., 2014*). Adult-specific *Tbx5* knockout mice demonstrate primary spontaneous and sustained AF, providing evidence supporting the genetic implication at this locus. GWAS have also implicated multiple genes involved in cardiomyocyte calcium handling, including *Atp2a2*, encoding the sarcolemmal calcium ATPase SERCA2, and *Sln* and *Pln*, encoding direct binding SERCA2 inhibitors sarcolipin and phospholamban, respectively. We have previously demonstrated that these cardiomyocyte calcium control genes are direct TBX5 targets (*Nadadur et al., 2016*). These observations suggested that tight transcriptional control of SERCA2 activity may be central to atrial rhythm robustness and that variation in SERCA2 expression and activity may contribute to AF risk.

The cellular mechanisms causing the irregular electrical activity in AF are believed to include an abnormal myocardial substrate and formation of an ectopic trigger. Abnormal substrate refers to altered electrical conduction between cardiomyocytes. Ectopic trigger refers to cardiomyocyte ectopy, or initiation of electrical activity at regions outside of the sinoatrial node. Both of these cellular phenomena are observed in *Tbx5* adult-specific mutant mice and have been associated with abnormal cellular calcium handling (*Dobrev, 2010*; *Voigt et al., 2012*; *Voigt et al., 2014*; *Vest et al., 2005*; *Shanmugam et al., 2011*; *Neef et al., 2010*; *Macquaide et al., 2015*; *Liang et al., 2008*; *Lenaerts et al., 2009*; *Hove-Madsen et al., 2004*; *Greiser et al., 2011*; *El-Armouche et al., 2006*; *Brundel et al., 1999*). We described the TBX5-dependent gene regulatory network essential for atrial rhythm control and identified downstream ion channels and transporters potentially important to rhythm control (*Nadadur et al., 2016*; *Yang et al., 2017*). Triggered activity in the form of early and delayed afterdepolarizations (EADs and DADs) observed in Tbx5-deficient atrial cardiomyocytes could be rescued by heavy buffering of cytoplasmic calcium (*Nadadur et al.,*

*2016*). *Tbx5*-dependent calcium handling has thereby emerged as a potential mediator of the myocardial physiologic abnormalities resulting in AF.

We sought to define the Tbx5-dependent cellular mechanisms responsible for abnormal calcium-dependent electrical activity. We found that *Tbx5*-dependent AF is associated with abnormal sarcoplasmic reticulum (SR) calcium uptake due to depressed SERCA2 expression, depressed SERCA function, and increased phospholamban expression. Decreased SR calcium uptake is compensated by increased $Ca^{2+}$ extrusion from cardiomyocytes via sodium-calcium exchanger (NCX) current ($I_{NCX}$), which provides a mechanism for TBX5-dependent action potential (AP) prolongation and the propensity for triggered cellular ectopy. In the setting of enhanced NCX mediated $Ca^{2+}$ efflux and depressed SR uptake, compensatory increases in L-type calcium current ($I_{CaL}$) balance calcium extrusion to maintain steady state calcium homeostasis. Together these calcium handling alterations contribute to AP prolongation and triggered activity.

We further demonstrated that calcium handling abnormalities, AP alterations, and triggered activity are all normalized by knockout of phospholamban, which prevents *Tbx5*-dependent AF. These results establish a direct link between depressed SR $Ca^{2+}$ sequestration, enhanced NCX activity, and AF. This model suggests that targeting calcium handling pathways may be a treatment approach for a subpopulation of AF patients.

## Results

### Cytoplasmic calcium is responsible for AP prolongation in *Tbx5*-mutant atrial cardiomyocytes

We previously reported that *Tbx5*-deficient atrial cardiomyocytes demonstrated AP prolongation and myocardial ectopy. We hypothesized that these defects were caused by cellular calcium handling abnormalities. We therefore surveyed the expression of known calcium handling genes in the adult-specific *Tbx5* knockout model. We assessed gene expression in *Tbx5^fl/fl^;R26^CreERT2^* and control *R26^CreERT2^* mice at 10 weeks of age following tamoxifen (TM) treatment at 8 weeks of age. Consistent with previous observations, the adult *Tbx5^fl/fl^;R26^CreERT2^* but not control mice developed spontaneous AF, showing an irregularly irregular heartbeat, by telemetric electrocardiogram (ECG) recordings (*Figure 1A,B*). As previously shown, APs and [Ca]_i transients were prolonged in *Tbx5^fl/fl^; R26^CreERT2^* (*Figure 1C*) (*Nadadur et al., 2016*). We assessed expression of genes important to cellular calcium handling in the left atrium by quantitative PCR (*Figure 1D*). mRNA transcripts for RyR2 (*Ryr2*) and SERCA2 (*Atp2a2*), two of the main determinants of sarcoplasmic reticulum (SR) calcium flux, were decreased by 61% and 71% respectively in *Tbx5^fl/fl^;R26^CreERT2^* mice compared to *R26^CreERT2^* controls (p=0.026 and p=0.001 for *Ryr2* and *Atp2a2* respectively) consistent with previous studies (*Nadadur et al., 2016*). In addition, phospholamban (*Pln*) mRNA expression was increased by 69% in *Tbx5^fl/fl^;R26^CreERT2^* compared to *R26^CreERT2^* (p=0.023), which would be expected to further depress SERCA2 activity. There was no significant difference in mRNA expression of the alpha 1C subunit of the L-type calcium channel (*Cacna1c*), the cardiac sodium calcium exchanger (*Ncx1*), or any of the calmodulins 1–3 (*Calm1, Calm2, Calm3*) (*Figure 1C*). These data are consistent with the hypothesis that the myocardial electrophysiology deficits in the *Tbx5*-deficient AF model may be due to abnormal calcium handling.

We examined the relationship between myocardial electrophysiology deficits and calcium flux in *Tbx5*-mutant atria. In steady state, with each cardiomyocyte contraction cycle, calcium entering the cardiomyocyte (L-type calcium channel, $I_{CaL}$) is extruded from the cell (predominantly via inward $I_{NCX}$). Similarly, calcium leaving the SR via RyR2 release or SR leak pathways is taken back up into the SR via SERCA2. We examined the effect of altered TBX5-dependent gene expression on these aspects of cardiomyocyte calcium flux. Given the observed changes in *Ryr2* and *Atp2a2* mRNA abundance, we hypothesized that AP prolongation in *Tbx5* deficient cardiomyocytes was due to calcium handling defects downstream of initial $Ca^{2+}$ entry through $I_{CaL}$. To test this, we recorded APs in the presence and absence of the L-type $Ca^{2+}$ channel blocker nifedipine. This approach blocks $Ca^{2+}$ entry into the cell and indirectly removes the effect of $Ca^{2+}$ entry on downstream $Ca^{2+}$ handling pathways, including SR $Ca^{2+}$ release/reuptake as well as the electrogenic effect of calcium transport out of the cell via inward $I_{NCX}$. 30 µM nifedipine completely inhibited L-type calcium current, preventing $Ca^{2+}$ entry or release of SR calcium in control *R26^CreERT2^* and *Tbx5^fl/fl^;R26^CreERT2^* (*Figure 2—*

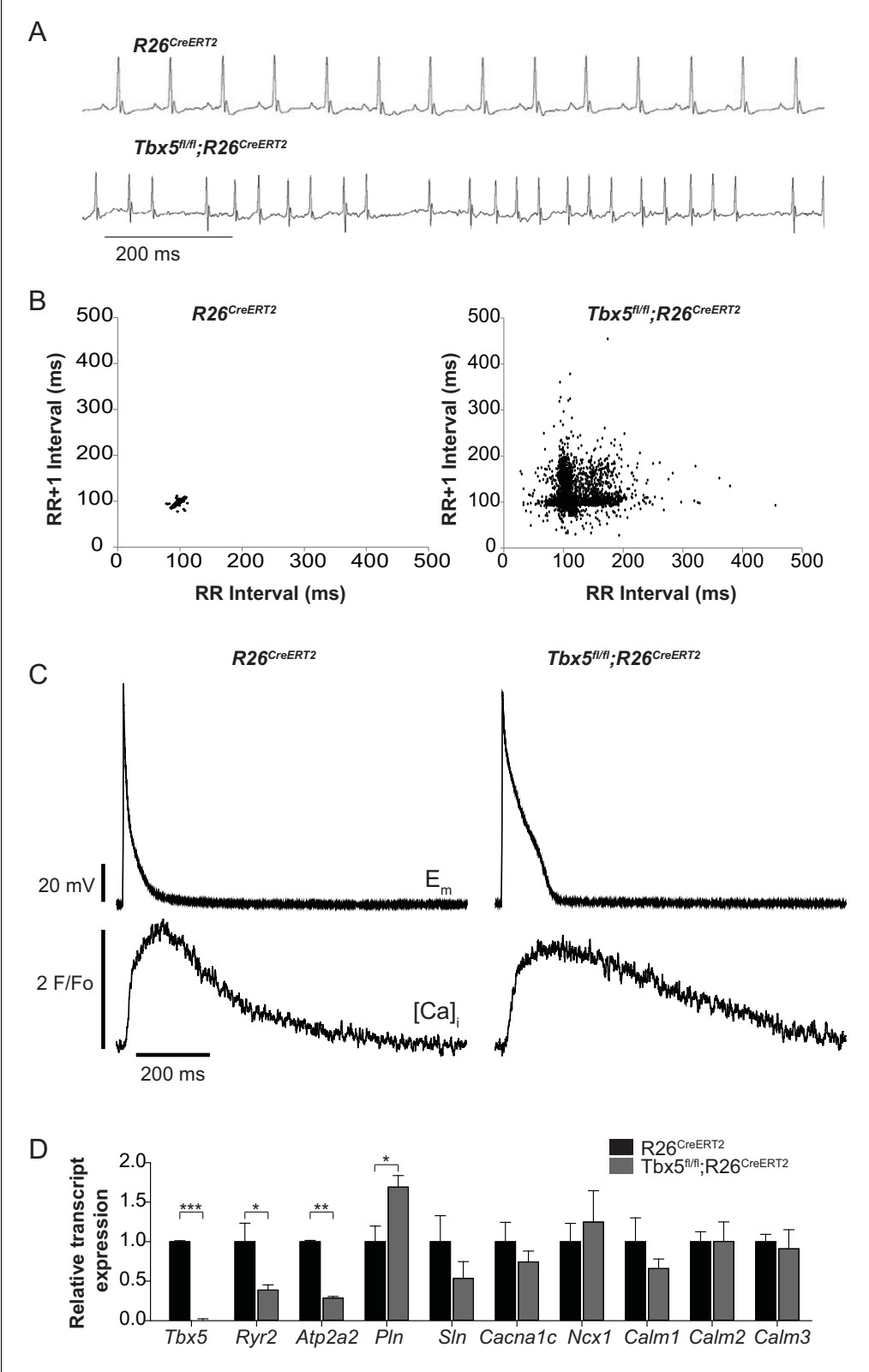

**Figure 1.** Atrial fibrillation in $Tbx5^{fl/fl};R26^{CreERT2}$ mice is associated with altered expression of genes important to cellular calcium handling. (**A**) $Tbx5^{fl/fl};R26^{CreERT2}$ mice developed spontaneous AF as assessed by surface ECG compared to $R26^{CreERT2}$. Traces are representative of 15 animals per genotype. (**B**) Poincaré plot shows irregularly irregular rhythm in $Tbx5^{fl/fl};R26^{CreERT2}$, consistent with AF, compared to normal sinus rhythm in $R26^{CreERT2}$ mice. Poincaré plots are each from one animal, andrepresentative of 15 animals per genotype. (**C**) Simultaneous AP and $[Ca]_i$ recordings show

*Figure 1 continued on next page*

*Figure 1 continued*

prolonged AP duration and slowed $[Ca^{2+}]_i$ transient decay in $Tbx5^{fl/fl};R26^{CreERT2}$ atrial cardiomyocytes compared to $R26^{CreERT2}$. Recordings are representative of simultaneous $[Ca]_i$ and $E_m$ recordings (myocytes/mice; dual $E_m$ and $[Ca]_i$ from 5/5 $R26^{CreERT2}$ and 17/5 $Tbx5^{fl/fl};R26^{CreERT2}$, $E_m$ only from 23/9 $R26^{CreERT2}$ and 20/9 $Tbx5^{fl/fl};R26^{CreERT2}$, and $[Ca]_i$ only from 27/6 $R26^{CreERT2}$ and 28/6 $Tbx5^{fl/fl};R26^{CreERT2}$). (D) Quantitative PCR was performed on RNA isolated from left atrial tissue of 3–5 animals per genotype. mRNA expression of a panel of calcium handling genes potentially important for rhythm regulation was determined. *Ryr2* and *Atp2a2* expression were decreased and *Pln* expression was increased in $Tbx5^{fl/fl};R26^{CreERT2}$ relative to $R26^{CreERT2}$ atria. (***p<0.001, **, p<0.01, *, p<0.05).

DOI: https://doi.org/10.7554/eLife.41814.003

The following source data and figure supplement are available for figure 1:

**Source data 1.** qPCR data for *Figure 1D*.
DOI: https://doi.org/10.7554/eLife.41814.005
**Figure supplement 1.** PCR primers.
DOI: https://doi.org/10.7554/eLife.41814.004

*figure supplement 1*). In control $R26^{CreERT2}$ atrial cardiomyocytes, the effect of nifedipine on AP duration (APD) was small, with 19 ± 4% shortening of APD at 90% repolarization (APD90) (p=0.008) (*Figure 2A*). However, in $Tbx5^{fl/fl};R26^{CreERT2}$ atrial cardiomyocytes, nifedipine had a profound effect: APD at 50% repolarization (APD50) was shortened by 16 ± 6% and APD90 by 61 ± 6% (p=0.02 and 0.007 respectively) (*Figure 2B,C*). Western blot with densitometry analysis for $Ca_V1.2$ showed no significant difference in protein expression (*Figure 2D*), in line with the qPCR data (*Figure 1D*), consistent with no TBX5-driven direct transcriptional regulation of L-type calcium channels. However, peak $I_{CaL}$ current was increased 92 ± 34% (p=0.027) in $Tbx5^{fl/fl};R26^{CreERT2}$ atrial cardiomyocytes compared to control $R26^{CreERT2}$ (*Figure 2E and F*). The inactivation kinetics at peak $I_{CaL}$ were accelerated $Tbx5^{fl/fl};R26^{CreERT2}$ compared to control $R26^{CreERT2}$ ( τ = 26.7 ± 3.4 ms vs. τ = 40.0 ± 3.0 ms; p=0.05). Steady-state $I_{CaL}$ inactivation was unchanged (*Figure 2—figure supplement 2*). These data suggest that increased $I_{CaL}$ may contribute to TBX5-loss associated AP prolongation and EADs. However, nifedipine also blocks SR $Ca^{2+}$ release as well as downstream $Ca^{2+}$ extrusion pathways, which also affect AP duration. Further, since late AP repolarization is dramatically prolonged (negative to −30 mV where $I_{CaL}$ is largely inactive) we hypothesized that *Tbx5*-deficiency disrupts $Ca^{2+}$ handling pathways downstream of $I_{CaL}$.

## Removal of *Tbx5* results in decreased $Ca^{2+}$ sparks and RyR2 expression, but no overall reduction in RyR2 open probability

Because RyR2 is a critically important sarcolemmal calcium extrusion channel and *Ryr2* mRNA was downregulated in *Tbx5*-mutant atria, we investigated the *Tbx5* dependent regulation of RyR2 protein expression and function. RyR2 protein expression was significantly decreased in left atria of $Tbx5^{fl/fl};R26^{CreERT2}$ mice compared to $R26^{CreERT2}$ mice by western blot (*Figure 3A*), consistent with the observed downregulation of *Ryr2* mRNA (*Figure 1B*). We hypothesized that decreased RyR2 contributed to abnormal $Ca^{2+}$ release from the SR and tested this by measuring local spontaneous RyR2-mediated $Ca^{2+}$ release events ($Ca^{2+}$ sparks) using confocal linescans (*Figure 3B*). The frequency of $Ca^{2+}$ sparks in $Tbx5^{fl/fl};R26^{CreERT2}$ atrial cardiomyocytes was decreased in comparison with $R26^{CreERT2}$ atrial cardiomyocytes at different pacing frequencies from 0 to 2 Hz (*Figure 3C*). A decrease in calcium sparks can be due to either decreased RyR2 open probability or a reduced SR calcium load. To differentiate these possibilities, we first examined RyR2 function in the setting of reduced RYR2 expression by performing a [3H]-ryanodine binding assay. [3H]-ryanodine binding to RyR2 correlates with RyR2 open probability (*Dobrev, 2010*). Despite reduced ryanodine receptor expression, overall ryanodine binding was unchanged over the majority of the physiological range of calcium values, with no shift in calcium sensitivity (*Figure 3D*). This observation suggests that the alterations in spark frequency were not due to changes in total RyR2 open probability. Instead, it may be caused by diminished SR $Ca^{2+}$ uptake, a SERCA-dependent property.

## Adult-specific *Tbx5* deficiency reduces SERCA activity and SR load while increasing sodium-calcium exchanger activity

We next focused on the balance of diastolic calcium efflux pathways as potential mediators of $Ca^{2+}$ mishandling by measuring SR $Ca^{2+}$ content and protein expression and function of SERCA2 and

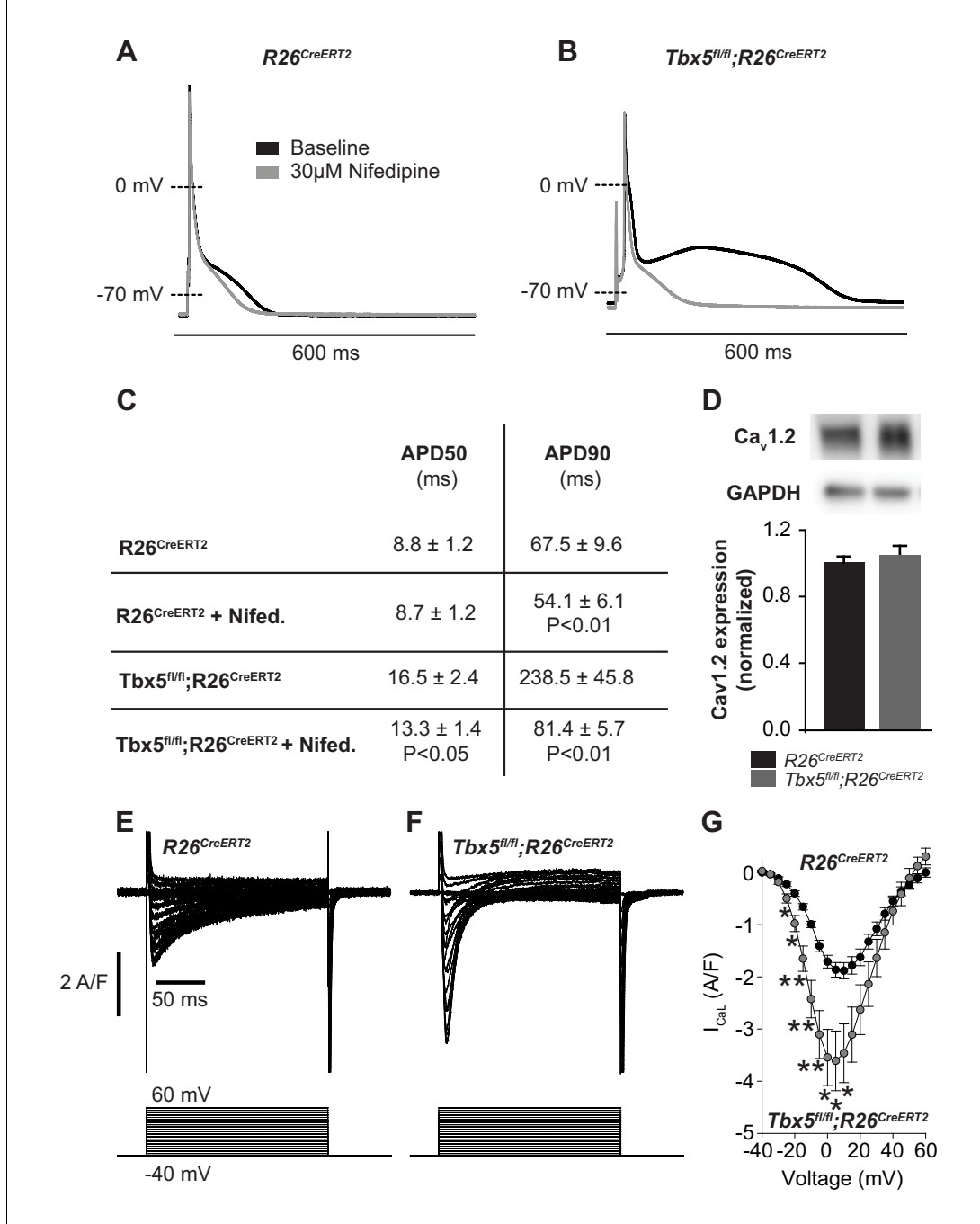

**Figure 2.** Calcium current blockade dramatically shortened the AP in $Tbx5^{fl/fl};R26^{CreERT2}$ atrial cardiomyocytes, consistent with the $[Ca]_i$ dependence of AP prolongation following TBX5 loss. (**A**) Representative recording of an AP from $R26^{CreERT2}$ atrial cardiomyocytes before and after 30 µM nifedipine treatment. (**B**) Representative recording of a $Tbx5^{fl/fl};R26^{CreERT2}$ atrial cardiomyocytes before and after nifedipine treatment. (**C**) Paired APD properties before and after treatment with 30 µM nifedipine (myocytes/mice; n = 8/3 $Tbx5^{fl/fl};R26^{CreERT2}$ and n = 6/4 $R26^{CreERT2}$). In $R26^{CreERT2}$ cardiomyocytes, the effect of nifedipine on APD90 was small, but significant 19 ± 4%. A much larger nifedipine effect was observed in $Tbx5^{fl/fl};R26^{CreERT2}$ cardiomyocytes. APD50 decreased by 16 ± 6% and APD90 decreased by 61 ± 6% in the presence of nifedipine. (**D**) Western blot of atrial tissue in five animals for each genotype showed protein expression for the alpha 1C subunit of the L-type calcium channel ($Ca_v1.2$) was unchanged. (normalized to GAPDH) (**E,F**) Representative $I_{CaL}$ recordings show Peak L-type calcium current was increased in $Tbx5^{fl/fl};R26^{CreERT2}$ cardiomyocytes compared to $R26^{CreERT2}$ (**G**) Average IV relationship of L-type calcium current (myocytes/mice; n = 22/7 $R26^{CreERT2}$ and 20/5 $Tbx5^{fl/fl};R26^{CreERT2}$). (\*\*\*p<0.001, \*\*, p<0.01, \*, p≤0.05).
DOI: https://doi.org/10.7554/eLife.41814.006

The following source data and figure supplements are available for figure 2:

**Source data 1.** AP and $I_{CaL}$ parameters for *Figure 2C,G*.

*Figure 2 continued on next page*

*Figure 2 continued*

DOI: https://doi.org/10.7554/eLife.41814.010

**Source data 2.** Western blot for *Figure 2D*.

DOI: https://doi.org/10.7554/eLife.41814.011

**Figure supplement 1.** 30 µM Nifedipine blocks L-type calcium current and calcium-induced calcium release in *R26^{CreERT2} and Tbx5^{fl/fl};R26^{CreERT2}* cardiomyocytes.

DOI: https://doi.org/10.7554/eLife.41814.007

**Figure supplement 2.** Steady state inactivation of $I_{CaL}$ was unchanged in *Tbx5^{fl/fl};R26^{CreERT2}* atrial cardiomyocytes.

DOI: https://doi.org/10.7554/eLife.41814.008

**Figure supplement 2—source data 1.** $I_{CaL}$ parameters for *Figure 2—figure supplement 2B*.

DOI: https://doi.org/10.7554/eLife.41814.009

NCX1. We observed that SERCA2 protein expression was decreased while NCX1 protein expression was increased in *Tbx5^{fl/fl};R26^{CreERT2}* in comparison with *R26^{CreERT2}* atria (*Figure 4A,B*). To define steady state SR $Ca^{2+}$ content, we loaded cardiomyocytes with Fluo-4 AM and paced with a train of field stimuli to achieve a steady state $Ca^{2+}$ content, peak $Ca^{2+}$ content, and rate of $Ca^{2+}$ removal were measured (*Figure 4C,D*). The $[Ca^{2+}]_i$ transient peaks were unchanged, but $[Ca^{2+}]_i$ transient decay rates, corresponding to SR $Ca^{2+}$ uptake and cellular $Ca^{2+}$ extrusion, were slowed in *Tbx5^{fl/fl};*

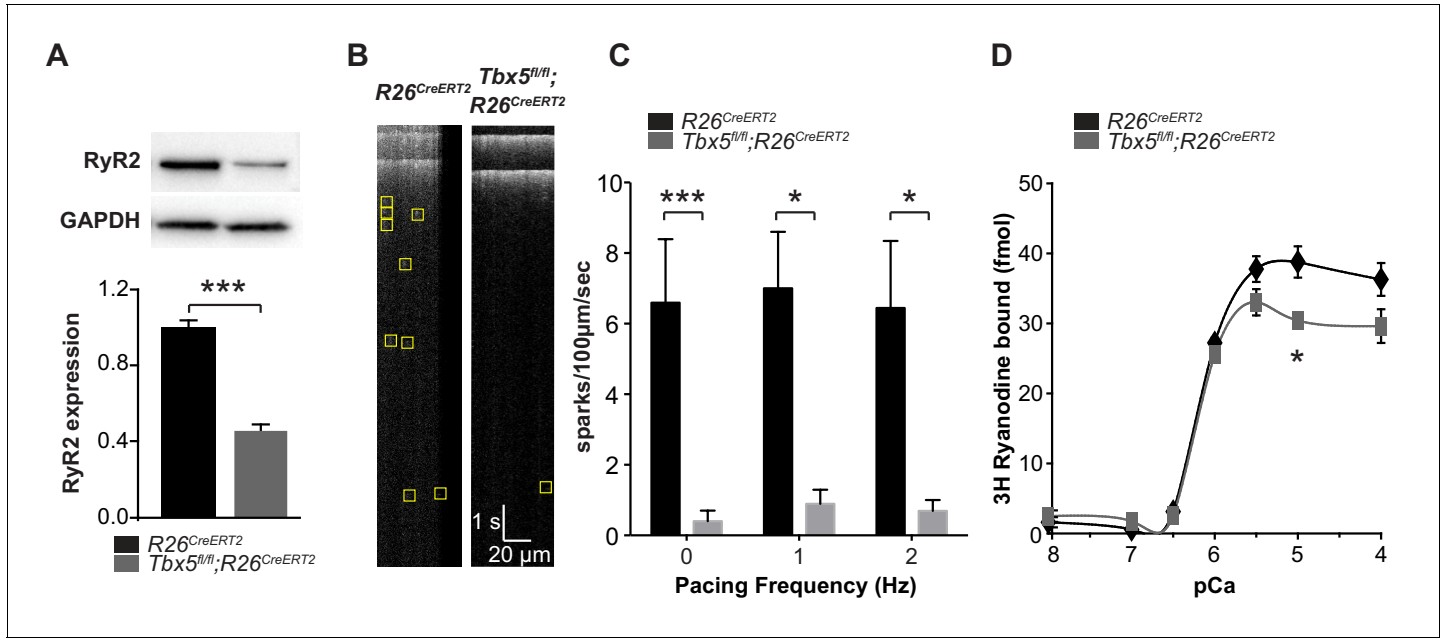

**Figure 3.** Spark frequency is reduced in *Tbx5^{fl/fl};R26^{CreERT2}* atrial cardiomyocytes. (A) Western blot from atrial tissue from 10 animals per genotype was used to measure RyR2 expression. RyR2 was significantly decreased in *Tbx5^{fl/fl};R26^{CreERT2}* atria compared to *R26^{CreERT2}* atria (normalized to GAPDH). (B) Fluo-4 loaded cardiomyocytes demonstrated reduced spark frequency in *Tbx5^{fl/fl};R26^{CreERT2}* compared to *R26^{CreERT2}* atrial cardiomyocytes (representative recordings). (C) Spark frequency was reduced at rest and after steady state pacing at different frequencies (myocytes/mice; n = 12/4 *Tbx5^{fl/fl};R26^{CreERT2}* and n = 12/3 *R26^{CreERT2}*). (D) Ryanodine binding assay (without normalization) demonstrated no significant difference over the physiologic range of $[Ca]_i$ in *Tbx5^{fl/fl};R26^{CreERT2}* compared to *R26^{CreERT2}* (*Weng et al., 2018*). Each measure corresponds to an assay performed on pooled atria from 8 to 10 mice with three independent measures per condition ($*p<0.05$, $**p<0.01$, $***p<0.001$).

DOI: https://doi.org/10.7554/eLife.41814.012

The following source data is available for figure 3:

**Source data 1.** Western blot for *Figure 3A*.

DOI: https://doi.org/10.7554/eLife.41814.013

**Source data 2.** Spark analysis for *Figure 3C*.

DOI: https://doi.org/10.7554/eLife.41814.014

**Source data 3.** Ryanodine binding assay for *Figure 3D*.

DOI: https://doi.org/10.7554/eLife.41814.015

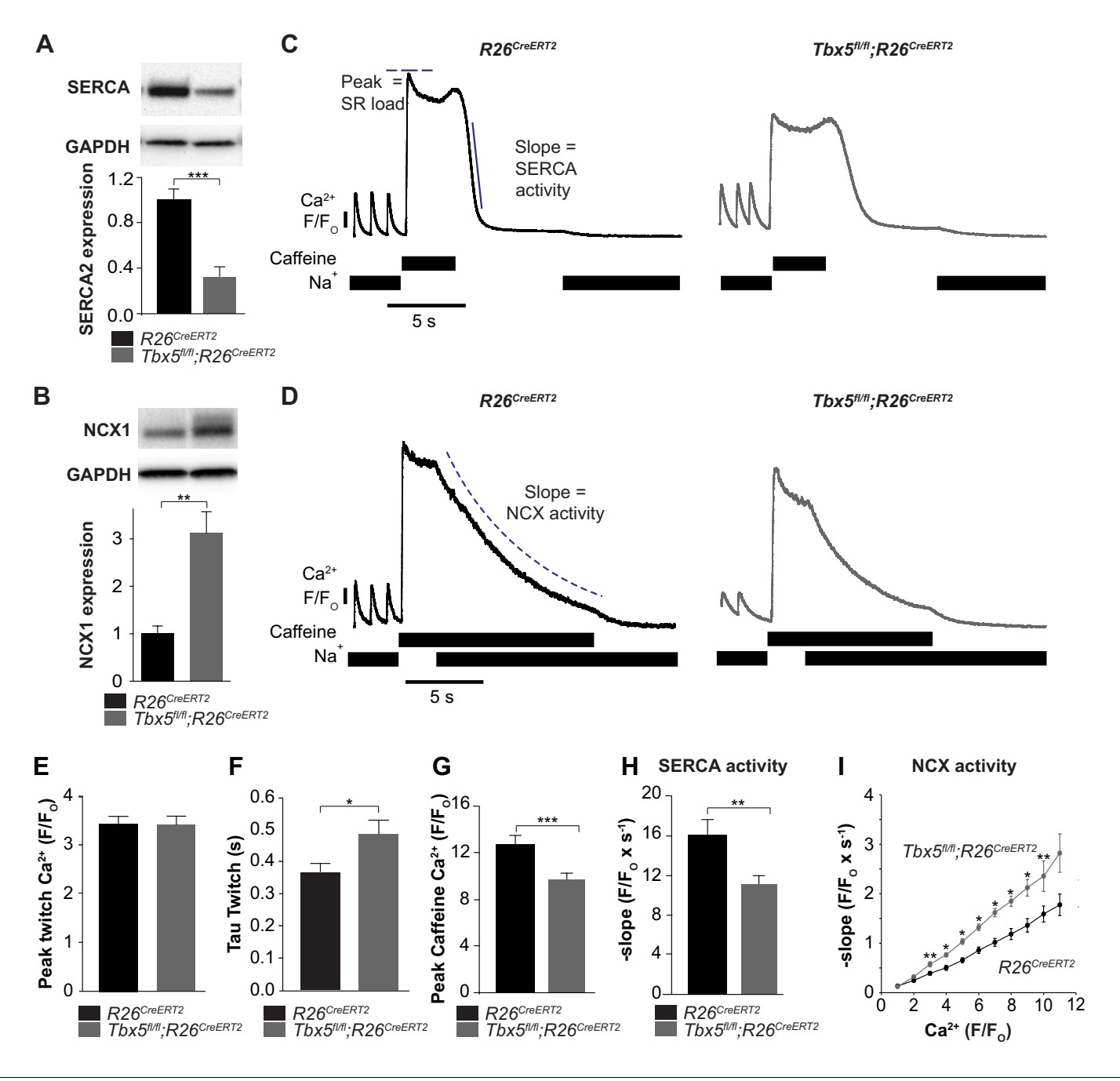

**Figure 4.** SERCA function is decreased while NCX function is increased in *Tbx5*$^{fl/fl}$;*R26*$^{CreERT2}$ atrial cardiomyocytes. (**A**) Expression of SERCA2 was significantly decreased (normalized to GAPDH) while (**B**) expression of NCX1 was significantly increased in *Tbx5*$^{fl/fl}$;*R26*$^{CreERT2}$ atria compared to *R26*$^{CreERT2}$ atria as measured by western blot in 10 animals per genotype. (normalized to GAPDH) (**C**) Application of Na$^+$ free caffeine solution after pacing to steady state at 1 Hz provided a measurement of SR load. In the absence of extracellular Na$^+$, [Ca$^{2+}$]$_i$ plateaued at high levels due to negligible role of non-NCX mediated extrusion in *R26*$^{CreERT2}$ and *Tbx5*$^{fl/fl}$;*R26*$^{CreERT2}$ atrial cardiomyocytes. Removal of caffeine in the absence of external Na$^+$ provided a measure of SERCA mediated SR calcium uptake (representative traces). (**D**) Restoration of external Na$^+$, in the presence of sustained extracellular caffeine provided a measure of NCX mediated calcium efflux (representative traces). (**E**) The peak of steady state twitch [Ca$^{2+}$]$_i$ transients was similar but (**F**) tau of [Ca$^{2+}$]$_i$ decay, determined from twitch [Ca$^{2+}$]$_i$ transients, was increased in *Tbx5*$^{fl/fl}$;*R26*$^{CreERT2}$ compared to *R26*$^{CreERT2}$ cardiomyocytes (myocytes/mice; n = 27/6 *R26*$^{CreERT2}$, n = 28/6 *Tbx5*$^{fl/fl}$;*R26*$^{CreERT2}$). (**G**) SR load, determined from peak caffeine transients was decreased in *Tbx5*$^{fl/fl}$;*R26*$^{CreERT2}$ compared to R26$^{CreERT2}$ cardiomyocytes (myocytes/mice; n = 34/6 *R26*$^{CreERT2}$, n = 32/6 *Tbx5*$^{fl/fl}$;*R26*$^{CreERT2}$). (**H**) SERCA activity, determined from the maximal rate of calcium decay was diminished in *Tbx5*$^{fl/fl}$;*R26*$^{CreERT2}$ compared to *R26*$^{CreERT2}$ cardiomyocytes (myocytes/mice;

*Figure 4 continued on next page*

*Figure 4 continued*

n = 29/3 $R26^{CreERT2}$, n = 32/3 $Tbx5^{fl/fl}$;$R26^{CreERT2}$). (I) NCX activity (decay slope), was increased at all levels of calcium in $Tbx5^{fl/fl}$;$R26^{CreERT2}$ cardiomyocytes (myocytes/mice; n = 35/3 $R26^{CreERT2}$, n = 21/3 $Tbx5^{fl/fl}$;$R26^{CreERT2}$). (*p<0.05, **p<0.01, ***p<0.001).

DOI: https://doi.org/10.7554/eLife.41814.016

The following source data and figure supplements are available for figure 4:

**Source data 1.** Western blot for *Figure 4A,B*.

DOI: https://doi.org/10.7554/eLife.41814.019

**Source data 2.** $Ca^{2+}$ handling parameters for *Figure 4E–H*.

DOI: https://doi.org/10.7554/eLife.41814.020

**Source data 3.** NCX activity for *Figure 4I*.

DOI: https://doi.org/10.7554/eLife.41814.021

**Figure supplement 1.** $[Ca^{2+}]_i$ transients recorded using 40 ms voltage clamp pulses demonstrate 23 ± 4% (p=0.02) reduction in peak calcium in $Tbx5^{fl/fl}$;$R26^{CreERT2}$ compared to $R26^{CreERT2}$ atrial cardiomyocytes.

DOI: https://doi.org/10.7554/eLife.41814.017

**Figure supplement 1—source data 1.** $Ca^{2+}$ transients for *Figure 4—figure supplement 1*.

DOI: https://doi.org/10.7554/eLife.41814.018

$R26^{CreERT2}$ compared to $R26^{CreERT2}$ atrial cardiomyocytes (*Figure 4E,F*) (*Nadadur et al., 2016*), consistent with defective $Ca^{2+}$ removal from the cytosol. We also measured $[Ca^{2+}]_i$ transients in voltage clamp mode using 40 ms square wave voltage clamp pulses from −80 to 0 mV (*Figure 4—figure supplement 1*). Similar to the field stimulation experiments, $[Ca^{2+}]_i$ transient decay rates were slowed, but $[Ca^{2+}]_i$ transient peaks were decreased by 23 ± 4% (p=0.02) in $Tbx5^{fl/fl}$;$R26^{CreERT2}$ cardiomyocytes compared to $R26^{CreERT2}$, which suggests that AP prolongation is essential to maintaining peak twitch $[Ca^{2+}]_i$. The latter experiment is also consistent with depressed SR loads in $Tbx5^{fl/fl}$;$R26^{CreERT2}$ compared to $R26^{CreERT2}$ atrial myocytes.

We hypothesized that decreased SERCA2 expression caused decreased SR load. We examined SERCA activity by synchronizing the opening of RyR2 channels while preventing $Ca^{2+}$ extrusion through NCX using caffeine containing, sodium-free, Tyrode solution. This provides a measurement of the maximum release of $Ca^{2+}$ into the cytosol from the SR, a measure of the SR $Ca^{2+}$ load (*Figure 4C,D*). SR $[Ca^{2+}]$ was reduced by 24 ± 8% (p=0.0005) in $Tbx5^{fl/fl}$;$R26^{CreERT2}$ compared with $R26^{CreERT2}$ atrial cardiomyocytes (*Figure 4G*). SERCA activity was assessed from $[Ca^{2+}]_i$ decay rate after SR release in the absence of external sodium (NCX inactive). Peak SERCA activity was reduced by 31 ± 9% (p=0.006) in $Tbx5^{fl/fl}$;$R26^{CreERT2}$ compared with $R26^{CreERT2}$ atrial cardiomyocytes (*Figure 4C,H*). NCX activity was assessed as the rate of change in $[Ca^{2+}]_i$ decay in $Na^+$ containing caffeine solution, preventing net SR uptake. Since NCX activity depends on $[Ca]_i$, we plotted NCX as a function of the $[Ca]_i$ signal. NCX activity was ~60% higher in $Tbx5^{fl/fl}$;$R26^{CreERT2}$ in comparison with $R26^{CreERT2}$ atrial cardiomyocytes (*Figure 4D,I*). Thus, removal of *Tbx5* causes decreased SR $Ca^{2+}$ load and decreased SERCA function, but increased NCX mediated $Ca^{2+}$ extrusion. Increased inward NCX activity promotes cardiomyocyte depolarization, providing a mechanism for prolonged APs and increased ectopy in *Tbx5*-mutant atrial cardiomyocytes.

## Genetic augmentation of SERCA activity and normalization of SR load eliminates susceptibility to AF

We hypothesized that *Tbx5* deficiency reduces SERCA activity by decreasing SERCA2 protein expression (*Figure 4A*) and increasing expression of phospholamban (*Pln*), a negative regulator of SERCA2 (*Figure 1D*). If these were the primary causes of decreased SERCA function in *Tbx5*-mutant atria, reduced PLN or PLN phosphorylation (relieving inhibition of SERCA2) would be expected to normalize SERCA function. Western blot analysis showed that PLN expression was significantly increased in $Tbx5^{fl/fl}$;$R26^{CreERT2}$ compared with $R26^{CreERT2}$ atria (*Figure 5A*). In addition, PLN phosphorylation was also increased at serine 16 in $Tbx5^{fl/fl}$;$R26^{CreERT2}$ compared to $R26^{CreERT2}$. These data suggest PLN phosphorylation may be a compensatory mechanism in response to decreased SERCA expression and activity, but is insufficient to normalize SERCA function (*Figure 4*). Thus, we hypothesized that reduction of *Pln* gene expression would be more effective in restoring SERCA function.

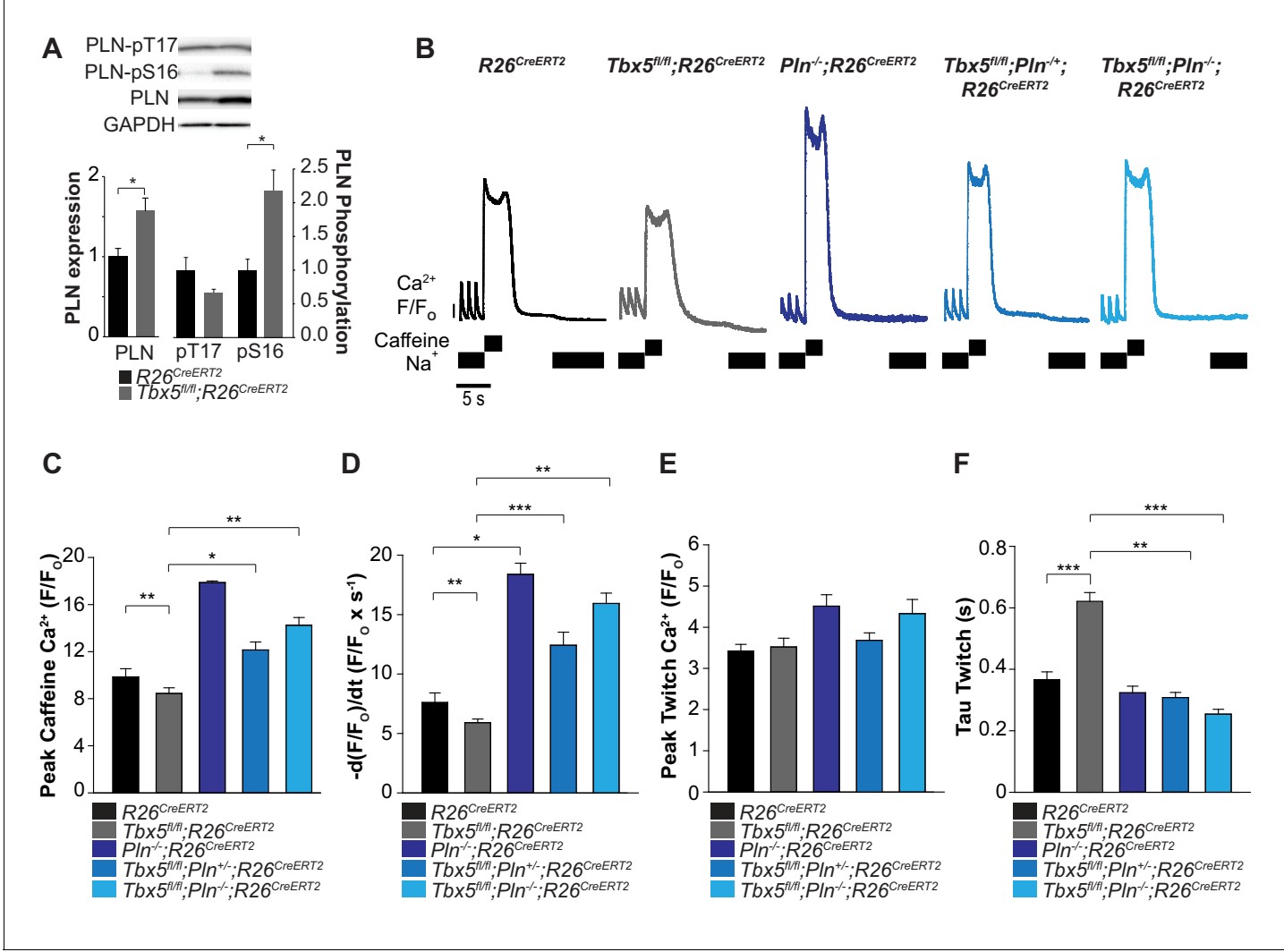

**Figure 5.** Phospholamban knockout normalized SERCA function in $Tbx5^{fl/fl};R26^{CreERT2}$. (A) PLN expression was increased in $Tbx5^{fl/fl};R26^{CreERT2}$ compared to $R26^{CreERT2}$ as measured by western blot with five animals per genotype. PLN expression was normalized to GAPDH. The proportion of PLN S16, but not T17 phosphorylation was also increased (normalized to PLN). (B) Representative SR load and SERCA measurements in $R26^{CreERT2}$, $Tbx5^{fl/fl};R26^{CreERT2}$, $Pln^{-/-};R26^{CreERT2}$, $Tbx5^{fl/fl};Pln^{-/+};R26^{CreERT2}$ and $Tbx5^{fl/fl};Pln^{-/-};R26^{CreERT2}$ atrial cardiomyocytes were collected as described in *Figure 4*. (C, D) SR load and SERCA function were significantly higher in $Tbx5^{fl/fl};Pln^{-/+};R26^{CreERT2}$ and $Tbx5^{fl/fl};Pln^{-/-};R26^{CreERT2}$ compared to $Tbx5^{fl/fl};R26^{CreERT2}$ cardiomyocytes and comparable to $R26^{CreERT2}$ cardiomyocytes. (E) $[Ca^{2+}]_i$ transient peaks were unchanged in $Tbx5^{fl/fl};R26^{CreERT2}$, $Tbx5^{fl/fl};Pln^{-/+};R26^{CreERT2}$ and $Tbx5^{fl/fl};Pln^{-/-};R26^{CreERT2}$, but increased in $Pln^{-/-};R26^{CreERT2}$ cardiomyocytes. (F) $[Ca^{2+}]_i$ transient decay rate in $Tbx5^{fl/fl};Pln^{-/+};R26^{CreERT2}$ and $Tbx5^{fl/fl};Pln^{-/-};R26^{CreERT2}$ cardiomyocytes were normalized to that of $R26^{CreERT2}$ cardiomyocytes (myocytes/mice; n = 34/3 $R26^{CreERT}$, n = 36/3 $Tbx5^{fl/fl};R26^{CreERT2}$, n = 30/3 $Pln^{-/-};R26^{CreERT2}$, n = 21/3 $Tbx5^{fl/fl};Pln^{-/+};R26^{CreERT2}$, n = 27/3 $Tbx5^{fl/fl};Pln^{-/-}$ atrial cardiomyocytes). (*p<0.05, **p<0.01, ***p<0.001).

DOI: https://doi.org/10.7554/eLife.41814.022

The following source data is available for figure 5:

**Source data 1.** Western blot for *Figure 5A*.

DOI: https://doi.org/10.7554/eLife.41814.023

**Source data 2.** $Ca^{2+}$ Handling parameters for *Figure 5C–F*.

DOI: https://doi.org/10.7554/eLife.41814.024

We assessed if *Pln* deficiency can affect SERCA function in a dose dependent manner by crossing the $Tbx5^{fl/fl};R26^{CreERT2}$ with germline *Pln* knockout mice ($Pln^{-/-};R26^{CreERT2}$) (*Luo et al., 1994*). We compared SR load and SERCA function in adult-specific *Tbx5; Pln* double mutant mice versus *Tbx5* mutant mice. We measured SR load and SERCA function using caffeine-induced SR release in atrial

cardiomyocytes from $R26^{CreERT2}$, $Tbx5^{fl/fl};R26^{CreERT2}$, $Pln^{-/-};R26^{CreERT2}$, $Tbx5^{fl/fl};Pln^{-/+};R26^{CreERT2}$ mice, and $Tbx5^{fl/fl};Pln^{-/-};R26^{CreERT2}$ mice. Control $Pln$ deficient mice ($Pln^{-/-};R26^{CreERT2}$) had increased steady state SR load and SERCA activity relative to $R26^{CreERT2}$ (*Figure 5C,D*). The decreased SR load and SERCA function observed in $Tbx5$ mutant mice ($Tbx5^{fl/fl};R26^{CreERT2}$) was converted to elevated SR load and SERCA function after the removal of $Pln$ ($Tbx5^{fl/fl};Pln^{-/-}; R26^{CreERT2}$) (*Figure 5C,D*). $Pln$ loss alone increased peak twitch calcium. However, in the setting of combined $Tbx5;Pln$ deficiency, peak twitch calcium and tau twitch were normalized to $R26^{CreERT2}$ values (*Figure 5E,F*).

We next tested the possibility that decreased SERCA function was the mechanism of TBX5-deficiency-driven AP prolongation and triggered activity and that decreased $Pln$ may rescue these defects. As we previously showed, $Tbx5^{fl/fl};R26^{CreERT2}$ atrial cardiomyocytes exhibited significantly prolonged APs and frequent EADs and DADs compared to $R26^{CreERT2}$ atrial cardiomyocytes (*Figure 6A,B*) (*Nadadur et al., 2016*). APs of $Pln^{-/-};R26^{CreERT2}$ atrial cardiomyocytes were similar to $R26^{CreERT2}$ controls (*Figure 6C*). The prolonged AP duration observed $Tbx5^{fl/fl};R26^{CreERT2}$ was rescued in both $Tbx5^{fl/fl};Pln^{+/-};R26^{CreERT2}$ and $Tbx5^{fl/fl};Pln^{-/-};R26^{CreERT2}$ atrial cardiomyocytes (43 ± 10% and 38 ± 5% shorter than $Tbx5^{fl/fl};R26^{CreERT2}$ respectively; p=0.01, 0.0007) (*Figure 6D,E*). Along with normalization of AP duration, we observed significantly fewer EADs and DADs in $Tbx5^{fl/fl};Pln^{-/-};R26^{CreERT2}$ cardiomyocytes (*Figure 6F*). The data demonstrate the importance of TBX5-driven SERCA activity on cellular electrophysiology and triggered activity in atrial cardiomyocytes and decreased $Pln$ rescues both SERCA function and cardiomyocyte electrophysiological abnormalities in $Tbx5$-mutant mice.

The data above show reducing $Pln$ gene dosage rescues calcium handling defects, AP prolongation and triggered activity observed in $Tbx5$-mutant atrial cardiomyocytes. We hypothesized that normalizing these cardiomyocyte cellular defects would reduce AF susceptibility in $Tbx5$ knockout mice (*Figure 7*). We performed intracardiac burst pacing. All $Tbx5^{fl/fl}; R26^{CreERT2}$ mice (6/6) paced into AF, compared to none of the $R26^{CreERT2}$ (0/5) or $Pln^{-/-};R26^{CreERT2}$ littermate controls (0/7). Consistent with our hypothesis, AF susceptibility was significantly decreased in $Tbx5; Pln$ compound knockouts: only 1/11 of $Tbx5^{fl/fl};Pln^{-/-};R26^{CreERT2}$ paced into AF (*Figure 7F*). Thus, $Tbx5$-deficiency induced AF is due to calcium handling abnormalities, specifically decreased SR load and SERCA activity, and that modulation of the SERCA2 inhibitor, $Pln$, normalized SERCA activity and AF susceptibility caused by $Tbx5$ loss.

## Discussion

AF initiation has been linked to calcium handling abnormalities in computational and in vivo disease models (*Grandi et al., 2011*; *Heijman et al., 2014*). Ectopic or triggered activity in the form of EADs and DADs are often due to calcium handling abnormalities that increase NCX activity and promote AF initiation (*Heijman et al., 2014*). However, the mechanism underlying genetic predispositions for AF remain poorly understood. Genetic variants and mutations at the $Tbx5$ locus are associated with increased risk for human AF and $Tbx5$-mutant mice show both spontaneous and burst pacing-induced AF (*Nadadur et al., 2016*). We report a calcium transport mechanism for $Tbx5$-dependent ectopic activity. We show that TBX5 is a critical regulator of SERCA-mediated SR calcium handling and that $Tbx5$-deficient mice have increased NCX-mediated $Ca^{2+}$ extrusion, balanced by increased $I_{CaL}$ mediated $Ca^{2+}$ influx. These calcium handling abnormalities provide a mechanism explaining the frequent triggered activity observed after TBX5 knockout. We show that decreasing phospholamban dosage can normalize TBX5-loss associated cellular calcium handling abnormalities, shorten AP duration, prevent triggered activity, and diminish AF susceptibility.

### Decreased SERCA activity and SR load in TBX5-loss associated atrial fibrillation

We and others have hypothesized that ectopic triggers of AF can be due to abnormal atrial calcium handling (*Greiser et al., 2011*; *Bers and Grandi, 2011*). Here we define this relationship in a model of spontaneous AF. We analyzed the major calcium transport pathways in atrial myocytes and demonstrated that the critical calcium handling deficit associated with $Tbx5$-loss is depressed SERCA-mediated SR calcium uptake. We report significant reduction of SERCA2 protein expression and function, consistent with human paroxysmal or chronic AF (*Voigt et al., 2012*; *Voigt et al., 2014*; *El-Armouche et al., 2006*; *Brundel et al., 1999*).

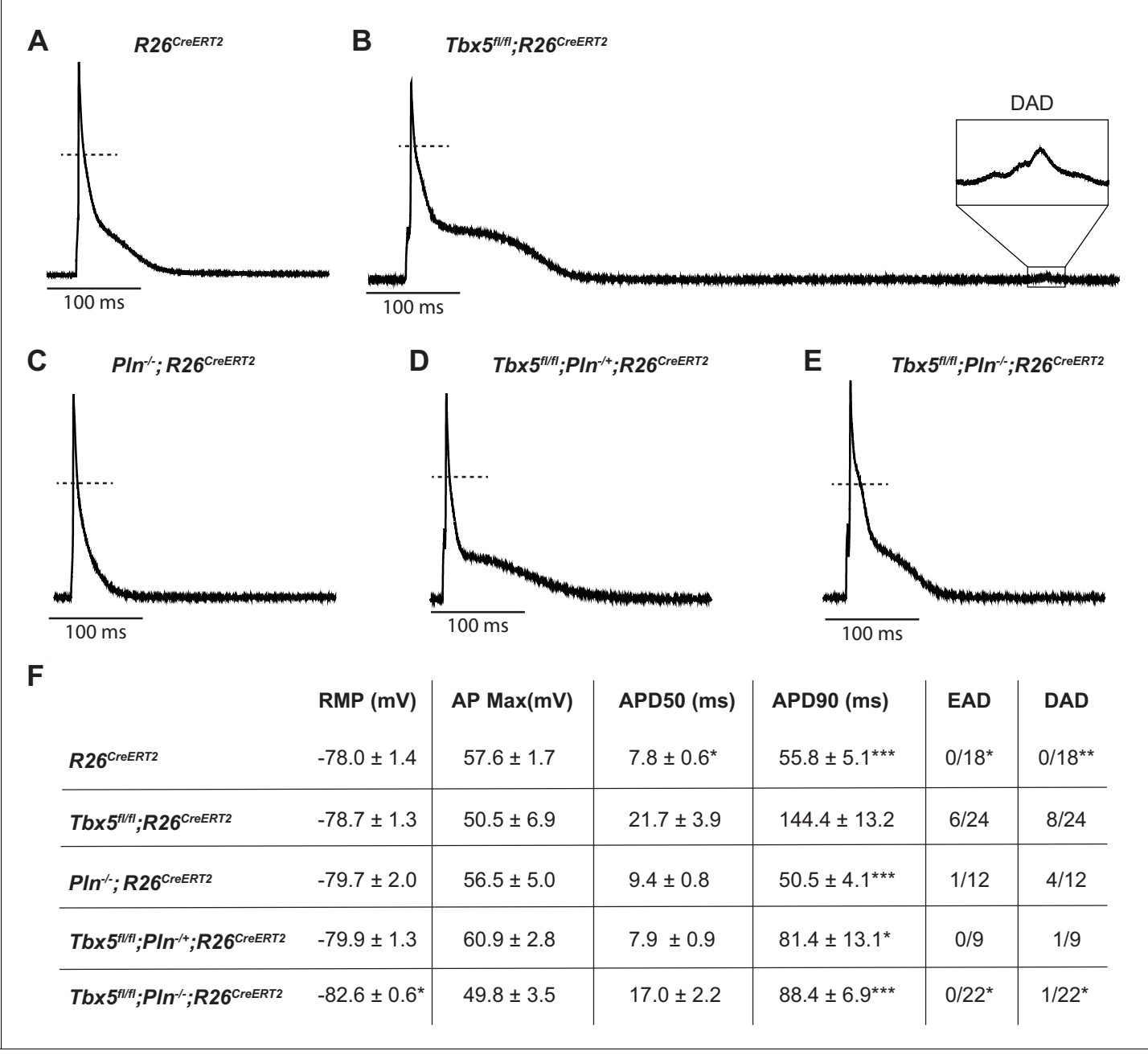

**Figure 6.** PLN knockout normalized AP duration and prevented triggered activity in *Tbx5^{fl/fl};R26^{CreERT2}*. Representative APs recorded from (**A**) *R26^{CreERT2}*, (**B**)*Tbx5^{fl/fl};R26^{CreERT2}*, (**C**)*Pln^{-/-};R26^{CreERT2}*, (**D**)*Tbx5^{fl/fl};Pln^{-/+};R26^{CreERT2}*, (**E**)*Tbx5^{fl/fl};Pln^{-/-};R26^{CreERT2}* atrial cardiomyocytes as described previously in *Figure 2*. (**F**) TBX5-loss dependent AP prolongation and frequency of triggered activity was normalized by phospholamban knockout (myocytes/mice: n = 18/7 *R26^{CreERT2}*, n = 24/12 *Tbx5^{fl/fl};R26^{CreERT2}*, n = 12/5 *Pln^{-/-};R26^{CreERT2}*, n = 9/3 *Tbx5^{fl/fl};Pln^{-/+};R26^{CreERT2}*, and n = 22/3 *Tbx5^{fl/fl};Pln^{-/-};R26^{CreERT2}*). (*p<0.05, **p<0.01, ***p<0.001).

DOI: https://doi.org/10.7554/eLife.41814.025

The following source data is available for figure 6:

**Source data 1.** AP Parameters for *Figure 6F*.

DOI: https://doi.org/10.7554/eLife.41814.026

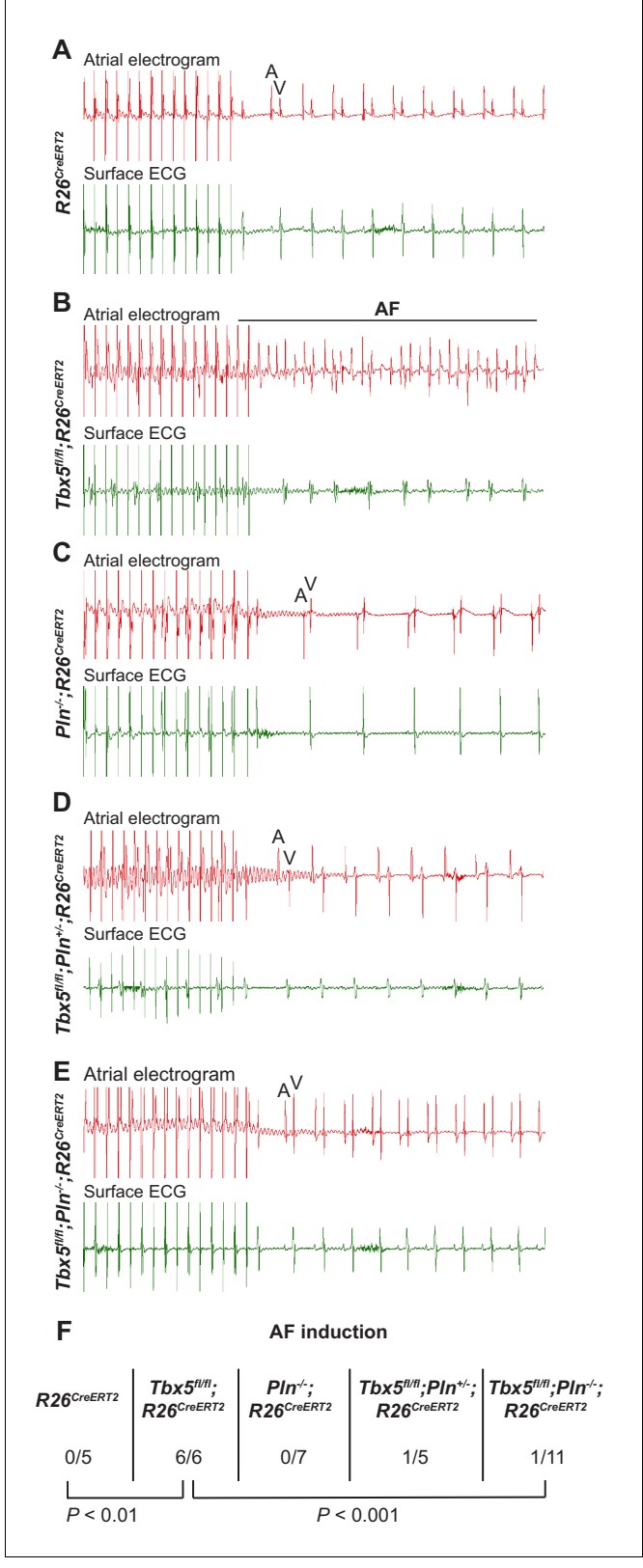

**Figure 7.** PLN deficiency protected against TBX5-loss associated AF Intra-atrial pacing was used to induce AF. Representative intracardiac atrial electrogram recordings and corresponding surface ECG are shown from (**A**) $R26^{CreERT2}$, (**B**)$Tbx5^{fl/fl};R26^{CreERT2}$, (**C**)$Pln^{-/-};R26^{CreERT2}$, (**D**)$Tbx5^{fl/fl};Pln^{-/+};R26^{CreERT2}$, (**E**)$Tbx5^{fl/fl};Pln^{-/-};R26^{CreERT2}$ atrial cardiomyocytes. A, atrial electrical signal; V, far field ventricular electrical signal. (**F**) AF was reproducibly

*Figure 7 continued on next page*

*Figure 7 continued*

demonstrated in 6/6 *Tbx5* knockouts in contrast to 1/11 *Pln/Tbx5* double knockouts, indicating rescue of atrial arrhythmogenesis. P values were determined by Fisher's exact test (n = 5 $R26^{CreERT2}$, n = 6 $Tbx5^{fl/fl};R26^{CreERT2}$, n = 7 $Pln^{-/-};R26^{CreERT2}$, and n = 5 $Tbx5^{fl/fl};Pln^{-/+};R26^{CreERT2}$, n = 11 $Tbx5^{fl/fl};Pln^{-/-}$ mice).

DOI: https://doi.org/10.7554/eLife.41814.027

The mechanism causing cardiomyocyte depolarizations from depressed SERCA activity must be indirect, given that SERCA2 is localized to the intracellular SR membrane and therefore does not directly contribute to membrane potential itself. Instead, slowed SR calcium uptake from depressed SERCA activity provides higher cytosolic calcium driving force for calcium extrusion from the cell via electrogenic inward $I_{NCX}$.

## Increased NCX activity in *Tbx5*- mutant atrial cardiomyocytes drives ectopic activity

We demonstrate increased NCX1 protein expression with *Tbx5* knockout, a finding also observed in human and other animal models of AF (*Neef et al., 2010*; *Lenaerts et al., 2009*; *Greiser et al., 2011*; *El-Armouche et al., 2006*). Since protein expression, electrochemical driving force, and allosteric calcium regulation can all affect amplitude of inward $I_{NCX}$ (*Blaustein and Lederer, 1999*), we measured NCX activity following loss of *Tbx5*. NCX activity was significantly increased at all levels of calcium (*Figure 4I*). Thus, increased NCX function coupled with prolonged $[Ca^{2+}]_i$ transients, drives increased inward $I_{NCX}$, providing additional depolarizing current during the AP, contributing to its prolongation. While increased NCX function may partially compensate for the depressed SERCA function to bring down calcium levels, it may also promote calcium-induced DADs in the setting of inappropriately timed SR calcium release events. Previous modeling in ventricular cardiomyocytes predicted countervailing functions of SERCA and NCX (*Li et al., 2011*), which we observed in *Tbx5* knockout mice. Our data further support that DADs, which are classically thought to relate to SR calcium overload can still occur with depressed SR loads in the appropriate context of depressed SERCA and elevated NCX function (*Voigt et al., 2012*).

Modeling suggests that compensatory increases in L-type calcium current in the setting of depressed SERCA function could be required to maintain systolic and diastolic calcium levels (*Li et al., 2011*). In line with the modeling, we observed enhanced peak $I_{CaL}$ with loss of *Tbx5*. This may account for early AP prolongation as well as EADs. It is interesting that peak $[Ca]_i$ is depressed using controlled square wave voltage clamp pulses (*Figure 4—figure supplement 1*), which suggests that 40 ms is insufficient to maintain calcium entry in *Tbx5* knockout, even in the setting of enhanced $I_{CaL}$. However, in the setting of AP prolongation (*Figure 2E–G*) peak twitch calcium levels are maintained. Additionally, the increase in calcium entering the cell through $I_{CaL}$ during the AP would be expected to balance a net increase in NCX mediated calcium extrusion (*Figure 4I*), a requirement for steady state $[Ca^{2+}]_i$ homeostasis. However, our observations that L-type calcium channel expression is TBX5-independent (*Figures 1D* and *2D*) and that genetically targeting only the $Ca^{2+}$ efflux pathways in our model is sufficient to restore normal electrical activity (*Figures 5–7*) suggest that the $I_{CaL}$ change is not a primary TBX5-dependent effect. Furthermore, although $I_{CaL}$ is increased, it quickly inactivates in TBX5 knockout cardiomyocytes (*Figure 2F*). Together with our observation that nifedipine normalizes the APD, this supports that enhanced calcium entry impacts APD via secondary $[Ca]_i$ dependent mechanisms.

## TBX5 loss results in reduced RyR2 expression

In addition to identifying the role of altered NCX and SERCA function, we assessed the importance of TBX5-driven RyR2 expression. *RYR2* is a known susceptibility locus for AF and *RYR2* mutations are correlated with AF (*Fatkin et al., 2017*; *Di Pino et al., 2014*). We observed that RyR2 protein expression was significantly depressed following *Tbx5* loss . Defective RyR2 function has also been associated with AF (*Vest et al., 2005*; *King et al., 2013*). Despite TBX5-dependent RyR2 expression, the ryanodine binding assay (*Xu et al., 1998*) suggested that RyR2 function is generally preserved over the physiologic range of calcium in *Tbx5*-mutant atria (*Figure 3D*). This suggests a compensatory mechanism must occur allowing for preserved RyR2 open probability in the setting of depressed

protein expression. For example, CaMKII is a potential regulator of RyR2 function which could increase the open probability and thereby increase steady leakage or favor spontaneous local $Ca^{2+}$ release events from the SR (*Vest et al., 2005*; *Neef et al., 2010*; *Fischer et al., 2015*) in *Tbx5*-deficient mice. In line with such compensation, we could not detect any differences in the calcium rise kinetics during controlled square wave voltage clamp pulses (*Figure 4—figure supplement 1*). Nevertheless, RyR2 compensation in the setting of reduced expression could contribute to abnormal triggered activity in the setting of *Tbx5* loss, and is an important topic for further investigation.

## Abnormalities of the TBX5/SERCA2/PLN regulatory axis drive AF formation

We found that depressed SERCA function in TBX5 knockout was completely normalized with heterozygous or homozygous phospholamban knockout, which normalized AP duration, decreased frequency of afterdepolarizations, and reduced AF inducibility. This finding demonstrates the importance of SERCA2 to the pathophysiology of AF in the *Tbx5*-loss model. While phospholamban has been associated with AF by GWAS (*Fatkin et al., 2017*; *Federico et al., 2017*), its functional role is less clear. Although PLN is predominantly found in the ventricle (*Bhupathy et al., 2007*), we showed not only its expression is increased in the atria in the context of *Tbx5* loss, but also PLN participates in rheostatic control of SERCA activity in the atria, which is sufficient to protect against AF inducibility (*Figures 5–7*). Our findings are further supported by patient studies. For example, in patients who experience post-operative AF, SERCA2 is significantly decreased in the atrial tissue (*Zaman et al., 2016*), but those with PLN mutations have decreased AF susceptibility in the context of arrhythmogenic right ventricular cardiomyopathy (*Bourfiss et al., 2016*). Thus, AF is a heterogeneous disease and there can be variability in how the calcium handling proteins are expressed (*Dai et al., 2016*) in different disease settings. The genetic background of an individual may be a critical determinant of how calcium handling moieties are disrupted to result in AF.

In summary, the most important features of the *Tbx5*-dependent SERCA2 and PLN regulatory axis are reduced SR uptake and load (*Figure 8*). In this setting, enhanced inward $I_{NCX}$ and $I_{CaL}$ contributes to AP prolongation, and, more importantly, to cardiomyocyte ectopy. Finally, we demonstrate PLN as a potential means to augment SERCA function, restoring normal atrial myocyte

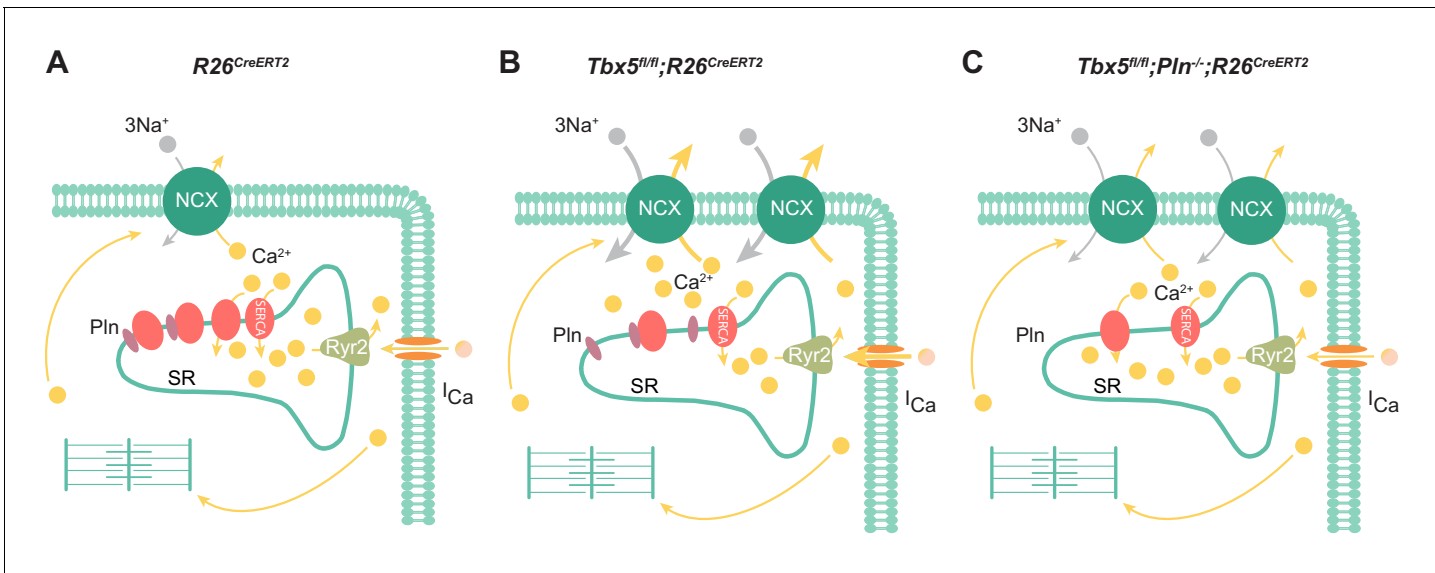

**Figure 8.** Model of TBX5-dependent calcium regulation in atrial cardiomyocytes. (**A**) Excitation-contraction coupling of atrial cardiomyocytes is achieved through regulation of intracellular calcium handling. (**B**) Adult-specific *Tbx5* knockout leads to decreased expression of SERCA2 and increased expression of PLN, leading to decreased SR $Ca^{2+}$ load. In addition, removal of *Tbx5* is associated with increased NCX1 expression and activity, thereby increasing $Ca^{2+}$ extrusion, which is balanced by increased L-type calcium entry. (**C**) Combined *Tbx5/Pln* knockout relieves repression of SERCA2. This results in normalization of SERCA activity and rescue of cardiomyocyte ectopy, triggered activity, and AF observed with *Tbx5* deficiency.
DOI: https://doi.org/10.7554/eLife.41814.028

electrical activity and normal sinus rhythm in *Tbx5* knockout mice. Thus, the *Tbx5* knockout model represents an excellent system to study pharmacologic rescue of SERCA activity, prevention of cardiomyocyte ectopy, and AF.

## Broader implications for clinical treatment of AF

AF has become an increasingly common cause of morbidity and mortality, underlying over one-third of stroke cases and significantly increases the risk for heart failure (*Nishida and Nattel, 2014*). Consequently, AF poses a significant socioeconomic burden. AF does not always exist in isolation, but rather in conjunction with other predisposing factors such as obesity, thyroid hormone alterations, or heart failure. Interestingly, disruptions in calcium handling proteins such as the SERCA2-PLN regulatory axis are implicated as predisposing factors. In AF compounded by heart failure, decreased SERCA2 and phosphorylated PLN, and increased NCX1 expression were observed (*Lugenbiel et al., 2015*). Decrease in phosphorylated PLN coupled with an increase in total PLN has been found in animal models of obesity, potentially increasing risk of AF (*Lima-Leopoldo et al., 2014*; *Lima-Leopoldo et al., 2008*). These findings suggest a need to evaluate an individual's genetic background as well as changes in calcium handling proteins when considering predisposing factors for AF.

Currently, there are few effective and targeted AF therapies, in part due to an incomplete understanding of the mechanisms underlying AF. Recent studies of specific genetic loci for AF susceptibility have opened new opportunities to identify specific mechanisms at play in subpopulations of AF patients. Understanding these specific mechanisms may facilitate more effective personalized therapies to target specific atrial $Ca^{2+}$ handling abnormalities. Our data is consistent with the knowledge that pharmacologic regulators of NCX1 or SERCA2 may normalize defects in cellular calcium handling in the atrium (*Dobrev, 2010*; *Jost et al., 2013*; *Nagy et al., 2014*; *Ferrandi et al., 2013*; *Parikh et al., 2012*). For example, a selective NCX1 inhibitor, ORM-10103, was shown to prevent cellular $Ca^{2+}$ handling abnormalities in ischemic ventricular cardiomyocytes, possibly by limiting calcium entry through outward $I_{NCX}$ (*Jost et al., 2013*; *Kormos et al., 2014*). The benefit of NCX inhibition might also be considered in human cases of AF with increased NCX activity. Further, resveratrol, which increases SERCA2 activity, has been shown to decrease AF, suggesting that targeting SERCA2 activity may be a viable therapeutic approach (*Bai et al., 2016*; *Chong et al., 2015*). In addition to providing specific insight into treating TBX5-loss associated AF, our findings may be more broadly applied. These data suggest that pharmacological treatment of AF may be improved by assessing for a defect in the TBX5-SERCA2-PLN axis followed by specifically targeting the defect to restore normal cardiomyocyte electrical activity. We expect this work and continued efforts to uncover mechanisms responsible for AF in subpopulations of patients will play a key role in advancing personalized therapeutics for AF.

# Materials and methods

**Key resources table**

| Reagent type (species) or resource | Designation | Source or reference | Identifiers | Additional information |
|---|---|---|---|---|
| Genetic reagent (M. musculus) | *Tbx5*$^{fl/fl}$ (*Tbx5*$^{tm1Jse}$) | PMID: 11572777, 27582060 | MGI:2387850 | Dr. Jonathan G Seidman (Harvard) |
| Genetic reagent (M. musculus) | *Pln*$^{-/-}$ (*Pln*$^{tm1Egk}$) | PMID: 8062415 | MGI:2158357 | Dr. Evangelia Kranias (University of Cincinnati) |
| Genetic reagent (M. musculus) | *Rosa26*$^{CreERT2}$(*Gt(ROSA)26 Sor*$^{tm1(cre/ERT2)Tyj}$) | PMID: 17251932, 27582060 | MGI:3790674 | Dr. Tyler Jacks (Massachusetts Insititute of Technology) |
| Antibody | Mouse anti-RyR2 | ThermoFisher | Cat. #: MA3-925 | WB (1:2000) |
| Antibody | Mouse anti-SERCA2 | ThermoFisher | Cat. #: MA3-919 | WB (1:1000) |
| Antibody | Mouse anti-NCX | ThermoFisher | Cat. #: MA3-926 | WB (1:1000) |
| Antibody | Mouse anti-PLN | Badrilla | Cat. #: A010-14 | WB (1:5000) |

*Continued on next page*

*Continued*

| Reagent type (species) or resource | Designation | Source or reference | Identifiers | Additional information |
|---|---|---|---|---|
| Antibody | Rabbit anti-pT17-PLN | Badrilla | Cat. #: A010-13 | WB (1:5000) |
| Antibody | Rabbit anti-pS16-PLN | Badrilla | Cat. #: A010-12 | WB (1:5000) |
| Antibody | Rabbit anti-Cav1.2 | Alomone | Cat. #: ACC-003 | WB (1:200) |
| Antibody | Mouse anti-GAPDH | Millipore | Cat. #: MAB374 | WB (1:10000) |
| Antibody | Goat anti-mouse-HRP | Thermofisher | Cat. #: 31437 | WB (1:5000) |
| Antibody | Goat anti-rabbit-HRP | Thermofisher | Cat. #: 31463 | WB (1:5000) |
| Chemical compound, drug | Fluo-4 AM | Thermofisher | Cat. #: 14201 | 10 µM x 20 min |
| Chemical compound, drug | Nifedipine | Sigma | Cat. #: N7634 | 30 µM |
| Chemical compound, drug | Collagenase Type 2 | Worthington Biochemical | Cat. # LS004177 | 1 g/L |
| Chemical compound, drug | Tamoxifen | MP Biomedicals | Cat#: 156738 | 2 mg/injection x three doses |
| Chemical compound, drug | Laminin | Invitrogen | Cat. #: 2039175 | 0.5 mg/ml |
| Chemical compound, drug | [3H]ryanodine | PerkinElmer | Cat. #: NET950250UC | |
| Chemical compound, drug | TRIzol | Invitrogen | Cat. #: 15596026 | |
| Chemical compound, drug | Ryanodine | MP Biomedicals | SKU #:0215377001 | |
| Chemical compound, drug | Caffeine | Sigma | Cat. #: C0750 | 10 mM |
| Software, algorithm | Clampex/Clampfit Data acquisition and analysis | Molecular Devices | Version 10.3.2.1 | |
| Software, algorithm | LabChart for electrophysiology studies | ADInstruments | Version 5 and 8 | |
| Software, algorithm | Buffering analyses using MaxChelator | Stanford | WEBMAXCLITE v1.15 | Chris Patton, Stanford University |
| Software, algorithm | Western Blot quantification ImageJ | NIH | Version 1.48 | |
| Software, algorithm | Hierarchical Statistical technique using R | R Core Team | Script from PMID: 29016722 | Ken Macleod, Imperial College London |
| Commercial Assay or Kit | qScript cDNA synthesis kit | Quanta | | |
| Commercial Assay or Kit | Power SYBR Green PCR Master Mix | Applied Biosystems | | |

## Generation of mice

The *Tbx5*<sup>fl/fl</sup>, *Pln*<sup>-/-</sup> and *Rosa26*<sup>CreERT2</sup> lines have all been previously described and were kept in a mixed genetic background (*Luo et al., 1994*; *Bruneau et al., 2001*; *Ventura et al., 2007*). Double knockout mice were generated by crossing *Tbx5*<sup>fl/fl</sup>;*R26*<sup>CreERT2</sup> mice with germline *Pln*<sup>-/-</sup> mice. After two generations, we obtained *Tbx5*<sup>fl/fl</sup>;*R26*<sup>CreERT2</sup> mice with either loss of one (*Pln*<sup>+/-</sup>) or both (*Pln*<sup>-/-</sup>) copies of *Pln*. All experiments were done using age- and genetic strain-matched littermate controls. Tamoxifen was administered for three consecutive days at a dose of 0.167 mg/kg body weight by intraperitoneal injection at 6–10 weeks of age, as previously described (*Nadadur et al., 2016*). All experiments were performed in accordance to The University of Chicago Institutional Animal Care and Use Committee (IACUC) approved protocol.

## ECG recordings

8-to 10- week-old mice were anesthetized using isoflurane, and telemetry transmitters (ETA-F10, Data Science International) were implanted in the back with leads tunneled to the right upper and left lower thorax, as previously described (Wheeler MT et al., JCI 2004). Baseline recordings were obtained for 24 hr after a post-implant recovery period of one day. ECG data was analyzed using LabChart 8 (AD Instruments).

## Intracardiac electrophysiology studies

Detailed protocols for intracardiac electrograms have been previously described (*Nadadur et al., 2016*). Briefly, 8- to 10- week-old mice were anesthetized with isoflurane and a vertical skin cutdown at the right jugular vein was performed. A 1.1 F octapolar catheter (EPR-800, Millar Instruments) was advanced in the right jugular vein to perform electrical stimulation. The catheter was connected to ADI BioAmp and PowerLab apparatus and signals were recorded using LabChart Software (ADInstruments). Atrial induction pacing was performed using burst pacing and the presence of at least three cycles of atrial tachycardia or fibrillation at least twice was considered positive.

## $[Ca^{2+}]_i$ transient measurement

Langendorff perfusion with 2 mg/mL of Collagenase Type 2 (Worthington Biochemical) at 5 ml/min was used to isolate atrial cardiomyocytes. Cardiomyocytes were then plated on laminin coated glass bottom dishes for 30 min prior to incubation with 10 μM Fluo-4/AM (Molecular Probes/Invitrogen) in normal Tyrode's solution containing (in mM): 140 NaCl, 4 KCl, 10 glucose, 10 HEPES, and 1 $MgCl_2$, 1 $CaCl_2$ pH 7.4 using NaOH for 20 min at room temperature. Cells were perfused with prewarmed Tyrode for 10 min prior to imaging. Imaging was performed on an Olympus microscope with a 20x objective lens, a LAMBDA DG-4 power source with 488 nm excitation and 515 nm emission filters and a PMT (photomultiplier tube) to record whole cell signal. Electrical field stimulation (Grass stimulator; Astro-Med) was performed at 1 Hz. SERCA and NCX measurements were performed by flowing sodium free Tyrode with 10 mM caffeine followed by sodium free Tyrode alone or Tyrode with caffeine respectively. Cells were returned to normal Tyrode in both cases at the end of the recording. $[Ca^{2+}]_i$ transients are presented as total fluorescence intensity normalized to resting fluorescence ($F/F_0$) obtained from steady-state resting conditions before field stimulation. $[Ca^{2+}]_i$ transients and sparks were acquired in line-scan mode (3 ms per scan; pixel size 0.12 μm) using a Zeiss confocal microscope.

## Whole-cell electrophysiological recordings

APs and voltage clamp recordings were recorded using standard ruptured patch protocol (*Nadadur et al., 2016*). We used current clamp mode with 0.5 nA ×2 ms current clamp pulses to measure APs. Voltage clamp mode was used to measure capacitance transients and to study $[Ca^{2+}]_i$ transients with fixed duration depolarizations. Cardiomyocytes are kept at 37°C and perfused with Tyrode solution (140 NaCl, 4 KCl, 1 $MgCl_2$, 1 $CaCl_2$, 10 HEPES, 10 Glucose, and pH 7.4 with NaOH). Internal pipette solution composition was (in mM): 20 KCl, 100 K-glutamate, 10 HEPES, 5 $MgCl_2$, 10 NaCl, 5 Mg-ATP, 0.3 Na-GTP. Patch pipettes (World Precision Instruments) were pulled to have a mean resistance of 3.5–5 MΩ. An Ag–AgCl pellet and 3M KCl agar bridge was used to ground the bath. Liquid junction potentials, were always corrected after cell rupture.

External solution for $I_{CaL}$ contained (in mM): 120 Tetraethylammonium-chloride, 10 CsCl, 10 Glucose, 10 HEPES, 1.5 $MgCl_2$, 1 $CaCl_2$, pH 7.4 with CsOH. Internal pipette solution contained (in mM): 100 Cs-methanesulfonate, 30 CsCl, 10 HEPES 5 EGTA, 2 $MgCl_2$, 5 Mg-ATP, pH 7.2 with CsOH. $I_{CaL}$ was recorded during 200 ms voltage clamp pulses from a holding potential of −40 mV to test potentials ranging from −40 to 60 mV, with pulses applied every 2 s in 5 mV increments. Peak current amplitudes were normalized to the cell capacitance ($C_m$) and presented as current density (A/F). Steady-state inactivation of $I_{CaL}$ was investigated using two-pulse protocol. Holding potential was −80 mV. The first pulse depolarized membrane from −60 to 20 mV with 10 mV increments during 500 ms, the second pulse depolarized the membrane to 10 mV for 50 ms. The inactivation curves were fit to a Boltzmann distribution.

Acquisition was performed using an Axopatch-200B amplifier connected to a Digidata1550A acquisition system (Axon Instruments, Foster City, CA, USA). In recording filtering at 2 kHz was

performed using the amplifier Bessel and sampled at 10 kHz. Analysis was performed using pCLAMP10 (Axon Instruments) and a home written analysis code.

## Western blots

Atrial tissue was collected and homogenized as described previously (*Alvarado et al., 2017*), in a buffer containing 0.9% NaCl, 10 mM Tris-HCl pH 6.8, 20 mM NaF and protease inhibitors. Equal amounts of protein, as determined by Bradford assay, were loaded. 50 µg of tissue homogenate, in Laemmli buffer, was separated by SDS-PAGE in 4–20% TGX or AnyKD precast gels (Bio-Rad). Proteins were transferred to PVDF membrane using the iblot2 transfer system (ThermoFisher) or wet transfer. Primary antibodies were as follows: anti-RyR2 (1:2000; MA3-925, ThermoFisher), SERCA2 (1:1000; MA3-919, ThermoFisher), NCX (1:1000; MA3-926, ThermoFisher), PLN (1:5000; A010-14, Badrilla), pT17-PLN (1:5000; A010-13, Badrilla), pS16-PLN (1:5000; A010-12, Badrilla), Cav1.2 (1:200; ACC-003, Alomone), GAPDH (1:10000; MAB374, Millipore). Secondary antibodies were: goat anti-mouse-HRP (1:5000; 31437, ThermoFisher) or goat anti-rabbit-HRP (1:5000; 31463, ThermoFisher). Secondary antibody concentrations were 5x higher when using the ibind Flex system. SuperSignal ECL reagent (ThermoFisher) was used to develop membranes followed by imaging with a ChemiDoc MP apparatus (Bio-Rad). Band intensities were quantified with the ImageLab software (Bio-Rad) or using ImageJ (NIH).

## [3H]Ryanodine binding assay

Binding assays were carried out following a protocol previously described (*Federico et al., 2017*). Binding mixtures contained 100 µg of protein from homogenates prepared from pooled atria (5–7 mice), 0.2 M KCl, 20 mM Hepes (pH 7.4), 6.5 nM [3H]ryanodine (PerkinElmer), 1 mM EGTA and enough $CaCl_2$ to set free $[Ca^{2+}]$ between 10 nM ($pCa^{2+}$ 8) and 100 µM ($pCa^{2+}$ 4). The ratio between $Ca^{2+}$ and EGTA was determined using MaxChelator (WEBMAXCLITE v1.15 http://maxchelator.stanford.edu/webmaxc/webmaxclite115.htm). Following a 2 hr incubation at 36°C, reactions were filtered through Whatman GF/B Filters using a Brandel M24-R Harvester. [3H]ryanodine binding was determined using a Beckman LS6500 scintillation counter and BioSafe II scintillation cocktail (RPI Corp). Non-specific binding was quantified in the presence of 2 µM unlabeled ryanodine (MP Biomedicals) and subtracted.

## Quantitative real time PCR

Left atrial tissue of *Tbx5fl/fl;R26CreERT2* and *R26CreERT2* mice was removed two weeks after receiving tamoxifen and RNA was isolated using a Trizol (Invitrogen) based method. Reverse transcription reaction was carried out using the qScript cDNA synthesis kit (Quanta) according to the manufacturer's protocol. Quantitative RT-PCR was performed using the Power SYBR Green PCR Master Mix (Applied Biosystems) and run on an Applied Biosystems AB7500 machine. Relative fold changes were calculated using the comparative threshold cycle method ($2^{-\Delta\Delta Ct}$), using glyceraldehyde-3-phosphate dehydrogenase (*Gapdh*) gene expression level as internal control.

## PCR primers

| Gene | F primer | R primer |
| --- | --- | --- |
| *Tbx5* | GGCATGGAAGGAATCAAGGT | CTAGGAAACATTCTCCTCCCTGC |
| *Ryr2* | CAAATCCTTCTGCTGCCAAG | CGAGGATGAGATCCAGTTCC |
| *Atp2a2* | CTGGTGATATAGTGGAAATTGCTG | GGTCAGGGACAGGGTCAGTA |
| *Pln* | TTATGCCAGGACGGCAAAAG | CACTGTGACGATCACCGAAG |
| *Sln* | CTGAGGTCCTTGGTAGCCTG | GGTGTGTCAGGCATTGTGAG |
| *Cacna1c* | CTACAGAAACCCATGTGAGCAT | CAGCCACGTTGTCAGTGTTG |
| *Ncx1* | TTCTCATACTCCTCGTCATCG | TTGAGGACACCTGTGGAGTG |
| *Calm1* | TGGGAATGGTTACATCAGTGC | CGCCATCAATATCTGCTTCTCT |
| *Calm2* | ACGGGGATGGGACAATAACAA | TGCTGCACTAATATAGCCATTGC |
| *Calm3* | GATGGCACCATTACCACCAAG | CGCTGTCTGTATCCTTCATCTTT |

## Statistical analysis

Values are represented as mean ±standard error of the mean (±SEM). Statistical significance for quantitative metrics of APs, SERCA, NCX, SR load, $I_{CaL}$, spark frequency, and $[Ca^{2+}]_i$ transients were determined using hierarchical statistical methods (*Sikkel et al., 2017*). Statistical significance for mRNA, and protein expression studies was determined using Student's t-test. Statistical significance of the nifedipine effect on AP duration was determined using two-tailed paired t-test. A two-tailed Fisher's exact test was used for statistical significance of count-based analysis of AF inducibility and EAD and DAD count. Statistical significance is designated as $*p < 0.05$, $**p < 0.01$, and $***p < 0.001$.

## Acknowledgements

We thank Evangalia Kranias for providing the phospholamban knockout mice (NIH HL26057). The project described was supported by R01 HL126509 and HL114010, AHA Collaborative Sciences Award, R33 HL123857, and T32HL007381.

## Additional information

### Funding

| Funder | Grant reference number | Author |
|---|---|---|
| National Institutes of Health | HL126509 | Ivan P Moskowitz |
| American Heart Association | Collaborative Sciences Award | Ivan P Moskowitz |
| National Institutes of Health | T32HL007381 | Wenli Dai |
| National Institutes of Health | HL114010 | Ivan P Moskowitz |
| National Institutes of Health | R33 HL123857 | Ivan P Moskowitz |
| National Institutes of Health | T32GM007281 | Wenli Dai |

The funders had no role in study design, data collection and interpretation, or the decision to submit the work for publication.

### Author contributions

Wenli Dai, Conceptualization, Resources, Data curation, Software, Formal analysis, Supervision, Funding acquisition, Validation, Investigation, Visualization, Methodology, Writing—original draft, Project administration, Writing—review and editing; Brigitte Laforest, Conceptualization, Data curation, Software, Formal analysis, Validation, Investigation, Visualization, Methodology, Writing—original draft, Project administration, Writing—review and editing; Leonid Tyan, Conceptualization, Data curation, Formal analysis, Investigation, Visualization, Writing—review and editing; Kaitlyn M Shen, Conceptualization, Data curation, Investigation, Visualization, Writing—review and editing; Rangarajan D Nadadur, Conceptualization, Data curation, Formal analysis, Funding acquisition, Investigation, Visualization, Methodology, Writing—review and editing; Francisco J Alvarado, Conceptualization, Software, Formal analysis, Investigation, Visualization, Writing—review and editing; Stefan R Mazurek, Conceptualization, Investigation, Methodology, Writing—review and editing; Sonja Lazarevic, Investigation, Methodology, Writing—review and editing; Margaret Gadek, Investigation, Writing—review and editing; Yitang Wang, Conceptualization, Data curation, Formal analysis, Writing—review and editing; Ye Li, Conceptualization, Data curation, Software, Formal analysis, Supervision, Investigation, Visualization, Methodology, Writing—review and editing; Hector H Valdivia, Conceptualization, Supervision, Investigation, Methodology, Writing—review and editing; Le Shen, Conceptualization, Formal analysis, Supervision, Investigation, Methodology, Project administration, Writing—review and editing; Michael T Broman, Conceptualization, Formal analysis, Supervision, Project administration, Writing—review and editing; Ivan P Moskowitz, Conceptualization, Resources, Data curation, Formal analysis, Supervision, Funding acquisition, Validation, Investigation, Visualization, Methodology, Writing—original draft, Project administration; Christopher R Weber, Conceptualization, Resources, Software, Supervision, Funding acquisition, Validation, Investigation,

Visualization, Methodology, Writing—original draft, Project administration, Writing—review and editing

## Author ORCIDs

Wenli Dai http://orcid.org/0000-0003-4328-9740
Brigitte Laforest http://orcid.org/0000-0001-6919-8922
Leonid Tyan http://orcid.org/0000-0001-9734-7286
Francisco J Alvarado http://orcid.org/0000-0003-0552-9029
Ye Li http://orcid.org/0000-0002-2969-2894
Le Shen http://orcid.org/0000-0002-4367-5476
Ivan P Moskowitz https://orcid.org/0000-0003-0014-4963
Christopher R Weber http://orcid.org/0000-0002-2117-3184

## Ethics

Animal experimentation: This study was performed in strict accordance with the recommendations in the Guide for the Care and Use of Laboratory Animals of the National Institutes of Health. All of the animals were handled according to approved institutional animal care and use committee (IACUC) protocols (#71737) of the University of Chicago. The protocol was approved by the Committee on the Ethics of Animal Experiments of the University of Chicago. All surgery was performed under isoflurane anesthesia, and every effort was made to minimize suffering.

## Decision letter and Author response

Decision letter https://doi.org/10.7554/eLife.41814.031
Author response https://doi.org/10.7554/eLife.41814.032

## Additional files

### Supplementary files

• Transparent reporting form

DOI: https://doi.org/10.7554/eLife.41814.030

### Data availability

All data generated or analysed during this study are included in the manuscript and supporting files.

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
