## [Decision Letter]

Thank you for submitting your article "A calcium transport mechanism for atrial fibrillation in Tbx5 mutant mice" for consideration by *eLife*. Your article has been reviewed by three peer reviewers, and the evaluation has been overseen by a Reviewing Editor and Richard Aldrich as the Senior Editor. The following individuals involved in review of your submission have agreed to reveal their identity: Robert S. Kass (Reviewer #1); David Eisner (Reviewer #3).

The reviewers have discussed the reviews with one another and the Reviewing Editor has drafted this decision to help you prepare a revised submission. Our goal is to provide the essential revision requirements as a single set of instructions, so that you have a clear view of the revisions that are necessary for us to publish your work.

Summary:

This is an interesting study of the effects of the T-box transcription factor TBX5 on the susceptibility to atrial fibrillation (AF), the most common and difficult to treat human cardiac arrhythmia. The authors provide convincing evidence that knockout of *Tbx5* in adult mice leads to (1) decreased expression of the SR Ca^2+^ pump SERCA2 and increased expression of phospholamban, leading to a decreased level of SR Ca^2+^ loading; and (2) increased Na-Ca exchanger (NCX1) expression and activity, providing a mechanism for prolonged action potentials and triggered activity. Additional knockout of phospholamban restored normal SR loading and action potential duration and prevented triggered activity and AF in *Tbx5*-knockout mice. These results provide a mechanistic link from genetic (GWAS) studies to the cellular mechanisms of AF.

Essential revisions:

The reviewers all found the paper to be interesting and important, but raised a number of concerns that must be adequately addressed before it can be accepted. Some of the required revisions will require further experimentation within the framework of the presented studies and techniques.

1) In Figure 2, the AP is lengthened in the *Tbx5* KO cells, and the effect of nifedipine to shorten the AP is greatly increased. The authors interpret this as an effect of prolonged and increased NCX current, but a change in Ca_v_1.2 activity may also contribute. Ca_v_1.2 channels could be more active in the KO cells if there is less Ca^2+^-dependent inactivation due to less CICR (from the reduced SR Ca content). This could contribute to the prolonged APD and to a more pronounced effect of nifedipine in blocking these channels. To address this point, the authors should conduct voltage clamp recordings of the Ca_v_1.2 current to monitor the degree of inactivation in WT vs. KO, with and without nifedipine. Simultaneous measurement of [Ca^2+^]_i_ would help establish a basis for these effects.

2) The authors suggest that a larger NCX current occurs in the *Tbx5* KO cells because of an increased NCX and decreased SERCA, but this needs some qualification. In the steady state, the amount of Ca pumped out of the cell must balance influx through L-type channels (assuming negligible PMCA activity). Thus, changes in SERCA and NCX cannot affect the integral of the NCX current. If the long plateau of the action potential in Figure 2B reflects an increase of delayed NCX current (due to prolongation of the Ca^2+^ transient by decreased SERCA), then this must be associated with a decrease of NCX in the earlier stage of the action potential. Is there any evidence for this? As suggested above (#1), simultaneous recording of the action potential and [Ca^2+^]_i_ in control and *Tbx5* KO cells would help to clarify this point.

3) In Figure 4E, the lack of effect of *Tbx5* KO on the amplitude of the twitch Ca^2+^ transient is interesting given that SR Ca^2+^ content is decreased. Two potential compensatory mechanisms should be addressed. 1) It may be a consequence of the prolongation of the AP. Stimulation of the cells with constant-duration square voltage clamp pulses while measuring [Ca^2+^]_i_ (see #1 above) would show whether the Ca transient is now less in *Tbx5* than control. 2) It looks as though, as a consequence of the slowing of decay of the Ca^2+^ transient (Figure 4F), the diastolic [Ca^2+^]_i_ is elevated. This may reduce cytoplasmic Ca^2+^ buffering such that a smaller total release of Ca^2+^ from the SR could result in an unchanged amplitude of the Ca^2+^ transient. Please comment on this possibility.

4) The statistical analysis may need to be modified. In Figure 1, please state the actual numbers of animals used in each group and for the t test. In Figure 2, 3C, 4E-I, and 5C-F, unless each myocyte comes from a different heart, it is invalid to use the number of cells as the "n" values for the t test (see Lazic, BMC Neurosci 2010;11:5). Either use hierarchical or nested statistical approaches (e.g., Sikkel, Francis and Howard, et al. 2017) or simply average cells from the same heart and reduce n to be the number of hearts.

5) Although PLB mRNA is increased in *Tbx5* KO mice, the functional contribution of PLB needs to be addressed by examining the phosphorylation status of PLB16 and PLB17.

6) The role of increased NCX in causing AF in *Tbx5*-KO mice needs to be tested directly. Can an NCX inhibitor eliminate EADs and DADs in *Tbx5*-KO myocytes?

7) The atrial phenotype of the *Tbx5* KO mouse (reduced RyR expression, increased NCX, reduced SERCA activity and SR Ca^2+^ load) are typical heart failure phenotypes. This raises the question of whether the observed phenotype indicates the underlying basis of AF, or is secondary to ventricular changes leading to heart failure. This question may be addressed using the mouse model before it has progressed to the heart failure stage.

---

## [Author Response]

Essential revisions:The reviewers all found the paper to be interesting and important, but raised a number of concerns that must be adequately addressed before it can be accepted. Some of the required revisions will require further experimentation within the framework of the presented studies and techniques.1) In Figure 2, the AP is lengthened in the Tbx5 KO cells, and the effect of nifedipine to shorten the AP is greatly increased. The authors interpret this as an effect of prolonged and increased NCX current, but a change in Ca_v_1.2 activity may also contribute. Ca_v_1.2 channels could be more active in the KO cells if there is less Ca^2+^-dependent inactivation due to less CICR (from the reduced SR Ca content). This could contribute to the prolonged APD and to a more pronounced effect of nifedipine in blocking these channels. To address this point, the authors should conduct voltage clamp recordings of the Ca_v_1.2 current to monitor the degree of inactivation in WT vs. KO, with and without nifedipine. Simultaneous measurement of [Ca^2+^]_i_ would help establish a basis for these effects.

We agree with the reviewers that I_CaL_ could contribute to APD prolongation, and that calcium influx and efflux must be balanced. To address this, we have measured I_CaL._ We find peak I_CaL_ is increased 92 ± 34% in *Tbx5^fl/fl^;R26^CreERT2^* cardiomyocytes in comparison to the *R26^CreERT2^* control. (new Figure 2E-G). Additionally, 30 μM nifedipine completely blocked I_Ca,L_ and eliminated any [Ca^2+^]_i_ transient in both TBX5 KO and control cardiomyocytes (new Figure 2—figure supplement 1). These data are consistent with a role of I_CaL_ in AP prolongation. Increased I_CaL_ may augment peak [Ca]_i_ in the setting of depressed SR load (Figure 4G) and also approximately balances the increase in NCX mediated calcium efflux (Figure 4I). We have added these observations to the Results and Discussion sections of the text.

2) The authors suggest that a larger NCX current occurs in the Tbx5 KO cells because of an increased NCX and decreased SERCA, but this needs some qualification. In the steady state, the amount of Ca pumped out of the cell must balance influx through L-type channels (assuming negligible PMCA activity). Thus, changes in SERCA and NCX cannot affect the integral of the NCX current. If the long plateau of the action potential in Figure 2B reflects an increase of delayed NCX current (due to prolongation of the Ca^2+^ transient by decreased SERCA), then this must be associated with a decrease of NCX in the earlier stage of the action potential. Is there any evidence for this? As suggested above (#1), simultaneous recording of the action potential and [Ca^2+^]_i_ in control and Tbx5 KO cells would help to clarify this point.

The reviewers correctly point out that calcium influx must match calcium efflux in steady state. We have now considered this point in the context of new experiments to determine I_CaL._ Our new finding that increased NCX current is offset by increased I_CaL_ now can explain steady state [Ca]_i_ balance. We have included these data (Figure 1 and 2) and include corresponding discussions. Nevertheless, the reviewers also point out that a decrease in NCX in the earlier stage of the action potential could also contribute to steady state [Ca]_i_ balance. Indeed, in the setting of AP prolongation, a decreased electrochemical drive for inward I_NCX_ could exist in the early part of the AP. Therefore, as suggested by the reviewers, simultaneous AP and [Ca^2+^]_i_ measurements were performed (new Figure 1C). Peak [Ca]_i_ was not depressed during the AP, consistent with our field stimulation experiments in Figure 4E. Thus, assuming that submembrane calcium in the vicinity of the NCX is also similar, then the electrochemical drive for inward I_NCX_ may be decreased in the early part of the AP in *Tbx5^fl/fl^;R26^CreERT2^* cardiomyocytes relative to *R26^CreERT2^* control. However, in the context of increased I_CaL_ and I_NCX_ activity, it is challenging to dissect the individual contributions of calcium-dependent currents at depolarized potentials. In the future we hope to better quantify the influx and efflux pathways for calcium during the AP with *Tbx5* deficiency using a combined experimental and computer modeling approach. A discussion of this point has been added to the manuscript.

3) In Figure 4E, the lack of effect of Tbx5 KO on the amplitude of the twitch Ca^2+^ transient is interesting given that SR Ca^2+^ content is decreased. Two potential compensatory mechanisms should be addressed. 1) It may be a consequence of the prolongation of the AP. Stimulation of the cells with constant-duration square voltage clamp pulses while measuring [Ca^2+^]_i_ (see #1 above) would show whether the Ca transient is now less in Tbx5 than control. 2) It looks as though, as a consequence of the slowing of decay of the Ca^2+^ transient (Figure 4F), the diastolic [Ca^2+^]_i_ is elevated. This may reduce cytoplasmic Ca^2+^ buffering such that a smaller total release of Ca^2+^ from the SR could result in an unchanged amplitude of the Ca^2+^ transient. Please comment on this possibility.

The new square wave experiment proposed by the reviewers is a very good idea because it allows us to assess the peak [Ca]_i_ without the effect of AP prolongation. The square wave experiments are now shown in Figure 4—figure supplement 1. In these experiments *Tbx5^fl/fl^;R26^CreERT2^* cardiomyocytes have a 23 ± 4% decrease in the calcium transient amplitude compared to controls. Our results suggest that AP duration (possibly coupled with increased I_CaL_ density) can explain the similarities in peak twitch calcium between *Tbx5^fl/fl^;R26^CreERT2^* and *R26^CreERT2^* cardiomyocytes(Figure 4E). It is also interesting to consider that altered buffering capacity could play a role. However, we did not detect any changes in rate of rise of the first 10 ms of the calcium transient as would be expected if a greater portion of calcium buffer was bound or if total calcium buffering capacity was altered. (*R26^CreERT2^*riseslope = 107 ± 11 F/F_o_*s; *Tbx5^fl/fl^;R26^CreERT2^*riseslope = 108 ± 18 F/F_o_*s, p=0.96). Furthermore, we want to emphasize that F_o_ was always recorded at rest without pacing, so measurements of SR load would be unaffected by this possibility. This new data has been added to the manuscript.

4) The statistical analysis may need to be modified. In Figure 1, please state the actual numbers of animals used in each group and for the t test. In Figure 2, 3C, 4E-I, and 5C-F, unless each myocyte comes from a different heart, it is invalid to use the number of cells as the "n" values for the t test (see Lazic, BMC Neurosci 2010;11:5). Either use hierarchical or nested statistical approaches (e.g., Sikkel, Francis and Howard, 2017) or simply average cells from the same heart and reduce n to be the number of hearts.

As suggested, we adjusted the statistical analysis using the hierarchical approach described in Sikkel et al [1]. Animal numbers are stated in the text where applicable and all of the figures are corrected to show significance indicators based on the revised analysis.

5) Although PLB mRNA is increased in Tbx5 KO mice, the functional contribution of PLB needs to be addressed by examining the phosphorylation status of PLB16 and PLB17.

We agree that the phosphorylation status of PLN is important. Thus, we performed western blot for PLN-S16 and PLN-T17 in *Tbx5^fl/fl^;R26^CreERT2^*and *R26^CreERT2^* control atria. Phosphorylation of PLN-S16 was significantly increased while that of PLN-T17 was unchanged in *Tbx5^fl/fl^;R26^CreERT2^*compared to *R26^CreERT2^* control (Figure 5a). Increased S16 phosphorylation may be a compensatory mechanism for the decrease in SERCA expression. However, based on measurements of SERCA expression and activity (Figure 4A, H), functional effects of PLN phosphorylation are not sufficient to normalize SERCA activity, but knockout of *Pln* is sufficient (Figure 5). This data and related discussion have been added to the manuscript.

6) The role of increased NCX in causing AF in Tbx5-KO mice needs to be tested directly. Can an NCX inhibitor eliminate EADs and DADs in Tbx5-KO myocytes?

As suggested by the reviewers, NCX inhibitors could theoretically reduce the frequency of DADs in our model. In our recordings using a new generation NCX inhibitor, ORM-10103, at 5μM, we observed a tendency towards action potential shortening (not significant) but no effect on frequency of triggered activity (EADs and DADs). A representative trace is shown in Author response image 1. However, very large I_NCX_ is still detectable in the presence of 5µM ORM-10103 using a square wave voltage clamp pulse to trigger SR release, as described in point 3 (Author response image 1). I_NCX_ inhibition was statistically insignificant (-4% ± 2%). Higher dosages of ORM-10103 or other NCX inhibitors are known to have off-target effects on Na^+^, Ca^2+^ and K^+^ channels, which would complicate interpretation [2,3], particularly in the setting of only partial NCX inhibition. Thus, it is unfortunately challenging to reach any conclusions from the NCX inhibitor data. As newer agents become available it would be reasonable to test their ability to inhibit triggered activity and AF in this model, and we address this in the Discussion.

Author response image 1>

**Author response image 1. respfig1:** ORM-10103 does not reduce the frequency of DADs. (**A**) Simultaneous E_m_/[Ca]_i_ recordings demonstrate prolonged APs and presence of DADs (one highlighted in inset). (**B**) Tail currents recorded after 40 ms square wave voltage clamp pulses reveal uninhibited I_NCX_.

7) The atrial phenotype of the Tbx5 KO mouse (reduced RyR expression, increased NCX, reduced SERCA activity and SR Ca^2+^ load) are typical heart failure phenotypes. This raises the question of whether the observed phenotype indicates the underlying basis of AF, or is secondary to ventricular changes leading to heart failure. This question may be addressed using the mouse model before it has progressed to the heart failure stage.

It is true that heart failure could complicate our interpretations. However, all experiments were performed at 2 weeks following tamoxifen injection. At this time point, the mice do not have any changes in ejection fraction, which begins after 3 weeks post-tamoxifen treatment [4].

References:

1) Sikkel MB, Francis DP, Howard J, et al. Hierarchical statistical techniques are necessary to draw reliable conclusions from analysis of isolated cardiomyocyte studies. Cardiovascular research. Dec 1 2017;113(14):1743-1752.

2) Jost N, Nagy N, Corici C, et al. ORM-10103, a novel specific inhibitor of the Na+/Ca^2+^ exchanger, decreases early and delayed afterdepolarizations in the canine heart. British journal of pharmacology. Oct 2013;170(4):768-778.

3) Amran MS, Homma N, Hashimoto K. Pharmacology of KB-R7943: a Na+-Ca^2+^ exchange inhibitor. Cardiovascular drug reviews. Winter 2003;21(4):255-276.

4) Nadadur RD, Broman MT, Boukens B, et al. Pitx2 modulates a Tbx5-dependent gene regulatory network to maintain atrial rhythm. Science translational medicine. Aug 31 2016;8(354):354ra115.